# Clinical Manifestations and Cytokine Profiles of the Th1, Th2, and Th17 Response Associated with SARS-CoV-2 Omicron Subvariants

**DOI:** 10.3390/biomedicines13092128

**Published:** 2025-08-31

**Authors:** Matheus Amorim Barreto, Amanda Mendes Silva Cruz, Delana Melo Volle, Wanderley Dias das Chagas Júnior, Iran Barros Costa, Juliana Abreu Lima Nunes, Aline Collares Pinheiro de Sousa, Izabel Keller Moreira Lima, Patrícia Yuri Nogami, Iami Raiol Borges, Luany Rafaele da Conceição Cruz, Paula Fabiane da Rocha Nobre, Edvaldo Tavares da Penha Junior, Jones Anderson Monteiro Siqueira, Victória Figueiredo Brito do Carmo, Darleise de Souza Oliveira, Hugo Reis Resque, Marcos Rogério Menezes da Costa, Rita Catarina Medeiros Sousa, Mirleide Cordeiro dos Santos, Maria Izabel de Jesus, Luana Soares Bargelata, Luciana Damascena da Silva, Igor Brasil-Costa

**Affiliations:** 1Laboratory of Immunology, Section of Virology, Evandro Chagas Institute, Health and Environment Surveillance Secretariat, Brazilian Ministry of Health, Ananindeua 66093-020, Brazil; irancosta@iec.gov.br (I.B.C.); patriciayuri.nogami@gmail.com (P.Y.N.); paula97fabiane@gmail.com (P.F.d.R.N.); darleiseoliveira@iec.gov.br (D.d.S.O.); 2Latent Cycle Virus Laboratory, Virology Section, Evandro Chagas Institute, Secretariat for Health and Environmental Surveillance, Brazilian Ministry of Health, Ananindeua 66093-020, Brazil; 3Virology Section, Evandro Chagas Institute, Secretariat for Health and Environmental Surveillance, Brazilian Ministry of Health, Ananindeua 66093-020, Brazil; amandamendes@iec.gov.br (A.M.S.C.); delanabezerra@iec.gov.br (D.M.V.); wanderleychagas@iec.gov.br (W.D.d.C.J.); jonessiqueira@iec.gov.br (J.A.M.S.); hugoresque@iec.gov.br (H.R.R.); mirleidesantos@iec.gov.br (M.C.d.S.); luanabarbagelata@iec.gov.br (L.S.B.); lucianasilva@iec.gov.br (L.D.d.S.); 4Evandro Chagas Institute, Health Ministry of Brazil, Ananindeua 67030-000, Brazil; julianaabreulima@hotmail.com (J.A.L.N.); izabelkmlima33@gmail.com (I.K.M.L.); iami.borges@unimedbelem.com.br (I.R.B.); enfluanycruz@gmail.com (L.R.d.C.C.); edvaldopenha@iec.gov.br (E.T.d.P.J.); vick.fig713@gmail.com (V.F.B.d.C.); 5Environment Section, Evandro Chagas Institute, Secretariat for Health and Environmental Surveillance, Brazilian Ministry of Health, Ananindeua 67030-000, Brazil; alinesousa@iec.gov.br (A.C.P.d.S.); mariajesus@iec.gov.br (M.I.d.J.); 6Belém UNIMED Hospital, Belém 66085-823, Brazil; marcos.costa@unimedbelem.com.br (M.R.M.d.C.); ritaclosset@uol.com.br (R.C.M.S.)

**Keywords:** COVID-19, cytokines, SARS-CoV-2 variants, interleukins

## Abstract

**Background:** The SARS-CoV-2 Omicron variant became the dominant driver during the COVID-19 pandemic due to its high transmissibility and immune escape potential. Although clinical outcomes are generally mild to moderate, the inflammatory mechanisms triggered by Omicron subvariants remain poorly defined. The goal of this study was to consider both viral evolution and the host immune response by assessing plasma cytokine levels in patients infected with SARS-CoV-2 Omicron subvariants. **Methods**: A total of 115 individuals were recruited, including 40 with laboratory-confirmed SARS-CoV-2 infection by RT-qPCR. Demographic, clinical, and comorbidity data were collected. Plasma levels of IL-6, TNF, IFN-γ, IL-4, IL-2, IL-10, and IL-17A were quantified using Cytometric Bead Array. Subvariant data were obtained from GISAID records and grouped into early (BA.1-lineage) and late (BA.4/BA.5-lineage) Omicron clusters. Statistical analysis included non-parametric and parametric tests, correlation matrices, and multivariate comparisons. **Results**: Pharyngitis, nasal discharge, cough, and headache were the most common symptoms among infected individuals. Despite no significant variation in symptom distribution across subvariants, infected patients showed higher levels of IFN-γ, TNF, IL-10, IL-4, and IL-2 compared to non-SARS-CoV-2 infected controls (*p* < 0.05). IL-4 and IL-10 levels were significantly higher in early Omicron infections. No associations were observed between cytokine levels and comorbidities. A significant correlation was found between reporting fewer symptoms and having received three vaccine doses. **Conclusions**: Infection with Omicron subvariants elicits a strong yet balanced cytokine response. Despite genetic divergence between lineages, immune and clinical patterns remain conserved, with vaccination appearing to mitigate the symptom burden.

## 1. Introduction

SARS-CoV-2 infection, the etiological agent of COVID-19, triggers a complex immune response that can be critical for both infection resolution and disease progression. Since the onset of the pandemic, it has become evident that the severity of COVID-19 depends not solely on viral replication but also on an exacerbated inflammatory response characterized by a dysregulated increase in inflammatory mediators, particularly pro-inflammatory cytokines, a phenomenon widely known as a cytokine storm [1].

In the context of viral infection, the activation of different T helper (Th) cell subsets plays a crucial role in modulating the immune response. Th1, Th2, and Th17 profiles, in particular, are especially relevant in COVID-19, as they are involved in coordinating inflammatory and antiviral responses, as well as regulating adaptive immunity [2,3,4]. The Th1 profile, associated with the production of IFN-γ, IL-2, and TNF, is linked to effective antiviral defense, whereas the Th2 profile, characterized by the production of IL-4 and IL-10, plays a more modulatory and anti-inflammatory role. In contrast, the Th17 response, represented by IL-17A, is essential for neutrophil recruitment and the maintenance of tissue inflammation [5].

An imbalance in the production of these cytokines may lead to more severe clinical outcomes, including the development of a cytokine storm, respiratory failure, and, in critical cases, multiple organ failure [6]. The severity of COVID-19 has been widely associated with dysregulation of the inflammatory response, characterized by elevated levels of various pro- and anti-inflammatory cytokines, which are frequently reported in hospitalized patients [7,8,9].

The emergence of multiple SARS-CoV-2 variants, largely driven by mutations in the Spike protein, as anticipated by early genomic studies, has significantly shaped the epidemiological trajectory of the COVID-19 pandemic [10,11]. Among the variants of concern (VOCs)—such as Gamma (P.1), Delta (B.1.617.2), and, more recently, Omicron (B.1.1.529)—the latter marked a turning point in the pandemic, characterized by substantial changes in viral transmissibility, immune evasion, and clinical presentation. Since its identification in late 2021, Omicron has rapidly diversified into multiple subvariants, prompting growing interest in its potential to differentially modulate host immune responses. This raises critical questions regarding how these subvariants influence the interplay between viral replication, immunological escape, and the host’s inflammatory response [10,12,13,14,15].

Understanding how these subvariants influence the balance between pro- and anti-inflammatory mediators is essential for elucidating the immunological mechanisms underlying disease severity and the variability in clinical manifestations. In particular, cytokines associated with Th1 (IFN-γ, IL-2, TNF), Th2 (IL-4, IL-10), and Th17 (IL-17A) responses, along with IL-6 as a central inflammatory marker, are key elements in defining the immune profile during acute SARS-CoV-2 infection.

Therefore, this study aimed to evaluate whether SARS-CoV-2 Omicron subvariants circulating in Northern Brazil were associated with differential expression of a targeted set of cytokines linked to Th1, Th2, and Th17 immune responses. Additionally, we investigated how these cytokine profiles relate to clinical characteristics, such as symptom burden, comorbidities, and specific subvariant clusters. By focusing on these key immunological pathways, we sought to describe relevant immuno-clinical associations observed in this regional cohort during the Omicron wave.

## 2. Materials and Methods

### 2.1. Data Collection and Biological Samples

This study included biological samples collected from 115 individuals (≥18 years), of both sexes, who sought care for suspected COVID-19 at two outpatient medical units located in the metropolitan region of Belém, Pará, Brazil. Sample collection was conducted between December 2021 and March 2022.

Following the signing of an informed consent form, clinical, sociodemographic, and comorbidity-related data were obtained through structured interviews. Simultaneously, 10 mL of whole blood was collected via venipuncture using a vacuum collection system containing ethylenediaminetetraacetic acid (EDTA) as an anticoagulant. In addition, nasopharyngeal swab samples were collected for SARS-CoV-2 molecular testing and genomic sequencing.

Samples were processed at the Immunology Laboratory of the Virology Section of the Evandro Chagas Institute. Whole blood was aliquoted and stored, and plasma and swabs were separated for subsequent analysis. Whole blood aliquots were used for RNA extraction and RT-qPCR, while plasma samples were reserved for cytokine quantification. Nasopharyngeal swabs were processed for viral RNA extraction and used in whole-genome sequencing procedures performed by a collaborating research group.

### 2.2. Genotyping Data Acquisition

The blood samples used in this study were previously processed by a collaborating research group responsible for SARS-CoV-2 variant identification. Viral RNA was extracted from nasopharyngeal swab specimens using a silica column-based commercial kit, followed by SARS-CoV-2 detection via RT-qPCR using probes and primers targeting conserved viral regions. For genotyping, whole-genome sequencing was carried out on the Illumina MiSeq platform, employing a library preparation protocol specific to SARS-CoV-2.

The genomic analysis and classification of SARS-CoV-2 lineages and sublineages were performed using the ViralFlow (version 1.3.0) workflow [16,17]. Raw sequencing reads underwent quality control and were aligned to the SARS-CoV-2 reference genome (NC_045512.2) as part of this pipeline. This workflow allows for automated processing of sequencing data, including genome assembly and lineage assignment. Consensus sequences were generated from high-quality reads and classified using the PANGOLIN tool. Clade assignment and sequence quality were further evaluated using the Nextclade platform.

The final lineage assignment for each individual was deposited in the GISAID (Global Initiative on Sharing All Influenza Data—https://gisaid.org/) database [18]. For the present study, variant data corresponding to the individuals included in our cohort were retrieved directly from GISAID using sample-specific identifiers.

### 2.3. Plasma Measurement of Cytokine Levels

Plasma levels of the cytokines IL-6, TNF, IFN-γ, IL-4, IL-2, IL-10, and IL-17A were quantified using flow cytometry, employing the BD FACS Canto II cytometer and the Human Th1/Th2/Th17 Cytometric Bead Array (CBA) kit (BD Biosciences, San Diego, CA, USA). The assay was performed strictly according to the manufacturer’s instructions.

Briefly, the method is based on the use of distinct sets of beads, each conjugated with a specific capture antibody targeting one of the selected cytokines. Upon incubation with the plasma samples and a PE-conjugated detection reagent, bead–cytokine–detection antibody complexes were formed and analyzed by flow cytometry.

Fluorescence signals were detected in the FL-3 channel, allowing for simultaneous and quantitative measurement of the cytokines in a multiplexed format. Data acquisition and analysis were conducted using BD FACSDiva™ software, and cytokine concentrations were calculated based on standard curves generated from known concentrations provided in the kit.

### 2.4. Statistical Analysis

All statistical analyses were conducted using JASP (version 0.17.2) and GraphPad Prism (version 10.5.0). Descriptive statistics were first applied to summarize clinical, demographic, and immunological data. The normality of data distributions was assessed using the Shapiro–Wilk test. Group comparisons were performed using Student’s *t*-test for normally distributed variables. For variables that deviated from normality, the Mann–Whitney U test was used. The Kruskal–Wallis test was applied to investigate associations between identified subvariants and cytokine levels. Associations between clinical characteristics, such as symptom occurrence and SARS-CoV-2 infection status or comorbidities, were evaluated using the Chi-Square test or Fisher’s exact test, depending on the sample size and distribution of responses. Spearman’s rank correlation was used to investigate monotonic relationships between plasma cytokine levels associated with Th1, Th2, and Th17 responses. This analysis was also applied to examine potential associations between symptom burden and cytokine expression. A significance level of *p* < 0.05 and a 95% confidence interval were adopted throughout.

### 2.5. Ethical Aspects

This study was approved by the Research Ethics Committee of the Evandro Chagas Institute (CAAE: 36869120.3.0000.0019; CEP approval number: 4.307.466) and conducted in accordance with Resolution No. 466/2012 of the Brazilian National Health Council (CNS) for research involving human subjects. All participants signed an informed consent form.

## 3. Results

### 3.1. Demographic Characterization and Diagnostic Confirmation

A total of 115 adult individuals, aged 18 years or older, participated in this study. Of these, 40 participants (34.78%) were diagnosed with COVID-19 through a positive molecular RT-qPCR test for SARS-CoV-2, while the remaining 75 individuals (65.22%) tested negative for this virus. Table 1 summarizes the main sociodemographic and clinical characteristics of the study population, including sex, age distribution, comorbidity status, and number of reported symptoms.

Participants were stratified into three age groups: <25, 25–50, and >50 years. Among SARS-CoV-2-positive individuals (*n* = 40), 5.0% were <25 years; 75.0% were 25–50 years, and 20.0% were >50 years. In the negative SARS-CoV-2 group (*n* = 75), 9.33% were <25 years; 70.67% were 25–50 years, and 20.0% were >50 years. No significant difference in age distribution was found between groups (*p* = 0.707, Chi-Square test). Regarding sex, 22.5% of positive individuals were male and 77.5% female, while in the negative SARS-CoV-2 group, 26.7% were male and 73.3% female. No significant differences in sex distribution were observed (*p* = 0.624, Chi-Square test).

Conversely, a statistically significant difference was observed in the prevalence of self-reported comorbidities. Among COVID-19 positive individuals, 40.0% (16/40) reported comorbidities, compared with 28.0% (21/75) in the non-SARS-CoV-2 infected group (*p* = 0.029, Chi-Square test).

### 3.2. Symptomatologic Analysis

The comparison between the groups with positive and negative molecular test results for SARS-CoV-2 revealed a statistically significant difference for some reported symptoms, as determined by the Chi-Square test (X^2^) in Table 1. Among individuals who tested positive, the most frequently reported symptoms were pharyngitis (82.5%), nasal discharge (80.0%), headache (77.5%), cough (72.5%), fatigue (57.5%), chills (57.5%), fever (55.0%), and muscle pain (52.5%). In contrast, the negative SARS-CoV-2 group exhibited lower symptom frequencies, particularly nasal discharge (56%), headache (53.3%), fatigue (36%), fever (46.7%), and cough (14.7%). After correction for multiple comparisons using the Benjamini–Hochberg false discovery rate (FDR) method, a statistically significant difference was observed between the groups only for chills (adjusted *p* = 0.014). Pharyngitis (before adjustment *p* = 0.017; adjusted *p* = 0.136) and painful breathing (before adjustment *p* = 0.041; adjusted *p* = 0.212), which were initially significant before correction, were not significant after adjustment. The other symptoms did not show statistically significant differences between the groups (*p* > 0.05).

Additionally, we evaluated the individual symptom burden based on the total number of symptoms reported per participant. The median number of reported symptoms was 7 (interquartile range: 5–9) in the positive group and 5 (IQR: 4–7) in the non-SARS-CoV-2 infected group. As the variable did not follow a normal distribution (Shapiro–Wilk test), the Mann–Whitney U test was applied and indicated a statistically significant difference between the groups (U = 541; *p* = 0.005), as illustrated in Figure 1A.

To further explore symptom expression within the SARS-CoV-2 positive group (*n* = 40), the distribution of 16 self-reported clinical symptoms was analyzed using the Chi-Square goodness-of-fit test to identify those occurring at frequencies significantly different from what would be expected under a uniform distribution. The most frequently reported symptoms in this group were pharyngitis (82.5%), nasal discharge (80.0%), headache (77.5%), cough (72.5%), fever (55.0%), chills (57.5%), fatigue (57.5%), muscle pain (52.5%), joint pain (37.5%), and loss of taste (25.0%). Less frequent symptoms included dyspnea (15.0%), anosmia (12.5%), abdominal pain (20.0%), vomiting (7.5%), and painful breathing (17.5%). Among the 16 symptoms assessed, four showed statistically significant frequencies (uncorrected *p* < 0.05), indicating higher-than-expected occurrences: pharyngitis (*p* = 0.017), cough (*p* = 0.008), nasal discharge (*p* = 0.014), and headache (*p* = 0.029). These results were derived from overall comparisons rather than pairwise analyses, and therefore, no post hoc correction for multiple testing was applied. These symptoms are highlighted in Figure 1B.

### 3.3. Omicron Subvariants Identified in the Sample

Genomic sequencing of SARS-CoV-2-positive samples revealed the exclusive circulation of subvariants belonging to the Omicron variant (B.1.1.529). A total of 11 distinct Omicron subvariants were identified, reflecting the high genetic diversification of this lineage during the study period. The most frequently detected subvariants were BA.1.14.1, identified in 30.0% of cases (*n* = 12); BA.1.1, in 25.0% (*n* = 10); BA.5.2.1, in 12.5% (*n* = 5); BA.5.1, in 10.0% (*n* = 4); and BA.1, in 7.5% (*n* = 3). The remaining subvariants—BA.1.1.14, BA.1.17, BA.4, BA.4.1, BA.5, and BA.5.1.22—were each identified in 2.5% of the sample (*n* = 1).

These proportions are illustrated in Figure 1C. The predominance of BA.1-derived subvariants suggests that early Omicron subvariants were still widely circulating during the sampling period, with a notable presence of BA.5-related subvariants at a lower frequency.

### 3.4. Correlation of the Most Frequent Symptoms of the Positive Group with the Identified Variants

To explore whether the occurrence of certain clinical symptoms in SARS-CoV-2 positive individuals differs according to Omicron subvariants, we conducted Fisher’s exact test to evaluate the relationship between each of the four most frequently reported symptoms—pharyngitis, nasal discharge, headache, and cough—and the 11 Omicron subvariants identified in the cohort (*n* = 40). The analysis did not reveal any statistically significant association between the presence of these symptoms and the infecting subvariant.

Pharyngitis was reported across all subvariants, with the highest frequencies in BA.1.14.1 (*n* = 10), BA.1.1 (*n* = 7), and BA.5.1 (*n* = 4), resulting in a *p*-value of 0.879 (Figure 1D); cough occurred most frequently in BA.1.1 (*n* = 8), BA.1.14.1 (*n* = 9), and BA.5.1 (*n* = 3), but no significant association was observed (*p* = 0.428) (Figure 1E); nasal discharge was present in nearly all subvariants, particularly BA.1.14.1 (*n* = 9), BA.1.1 (*n* = 6), and BA.5.2.1 (*n* = 4), with a *p*-value of 0.904 (Figure 1F); headache showed a broad distribution among subvariants, especially in BA.1.14.1 (*n* = 9), BA.1.1 (*n* = 7), and BA.5.2.1 (*n* = 3), with a *p*-value of 0.981 (Figure 1G).

These results suggest that the elevated frequencies of these four symptoms within the SARS-CoV-2-positive group cannot be explained by the subvariant distribution in this sample. Rather, they appear to represent common clinical features of acute Omicron infection, irrespective of subvariants

### 3.5. Comparison of Plasma Cytokine Levels Between SARS-CoV-2-Positive and -Negative Individuals

To investigate potential differences in inflammatory profiles, plasma concentrations of IL-17A, IFN-γ, TNF, IL-10, IL-6, IL-4, and IL-2 were compared between individuals tested for SARS-CoV-2 by RT-qPCR. As all cytokine variables deviated from normality according to the Shapiro–Wilk test (*p* < 0.001 for all markers), the Mann–Whitney U test was applied for group comparisons. Significant differences were identified in the levels of IFN-γ (*p* = 0.020), TNF (*p* < 0.001), IL-10 (*p* < 0.001), IL-4 (*p* < 0.001), and IL-2 (*p* < 0.001), all of which were elevated in the SARS-CoV-2-positive group. Conversely, no statistically significant differences were found for IL-17A (*p* = 0.311) or IL-6 (*p* = 0.673). These findings are illustrated in Figure 2A–G, which displays raincloud plots representing the distribution, density, and central tendency of cytokine levels across the two study groups.

### 3.6. Association Between Clinical Symptoms and Plasma Cytokine Levels in Patients with Acute COVID-19

To investigate whether specific clinical symptoms were associated with alterations in cytokine levels, we conducted an association analysis between the 16 self-reported symptoms and the plasma concentrations of seven cytokines: IL-17A, IFN-γ, TNF, IL-10, IL-6, IL-2, and IL-4, in individuals who tested positive for SARS-CoV-2. The corresponding *p*-values for each cytokine-symptom association are presented in Appendix A.

Among the comparisons performed, two nominal *p*-values were below the conventional significance threshold: the association between taste loss and IFN-γ levels (*p* = 0.038) and abdominal pain and TNF levels (*p* = 0.026). However, these results did not remain significant after correction for multiple comparisons using the Benjamini–Hochberg method (adjusted *p* = 0.962 for both). Post-correction *p*-values are presented in Appendix A.

All other symptoms, including pharyngitis, headache, fever, dyspnea, myalgia, and others, showed no statistically significant associations with any of the measured cytokines. These findings suggest that in this cohort, individual clinical manifestations were not significantly associated with circulating levels of Th1-, Th2-, or Th17-associated cytokines, nor with the central inflammatory marker IL-6.

### 3.7. Correlation Between Cytokine Levels and Identified Omicron Subvariants

Patients were categorized according to the Omicron subvariant with which they were infected, as identified by genomic sequencing. Plasma concentrations of various pro-inflammatory and regulatory cytokines, including IL-6, TNF, IL-4, IFN-γ, IL-17A, IL-10, and IL-2, were quantified in pg/mL. Due to the frequently non-parametric nature of cytokine data, the Kruskal–Wallis test was employed to perform comparisons between the subvariant groups, ensuring the statistical robustness of the analysis.

The results, detailed in Figure 3A–G of the report, consistently demonstrated the absence of statistically significant differences in the levels of all evaluated cytokines among the Omicron subvariant groups. The obtained *p*-values were as follows: for IL-6, *p* = 0.095; for TNF, *p* = 0.241; for IL-4, *p* = 0.154; for IFN-γ, *p* = 0.278; for IL-17A, *p* = 0.725; for IL-10, *p* = 0.374; and for IL-2, *p* = 0.397. The uniformity of these *p*-values, all exceeding the significance threshold of 0.05, suggests that the innate and adaptive immune response mediated by these specific cytokines, within this cohort, was not differentially modulated by the specific Omicron subvariants. This finding may indicate that for these particular cytokines, the host’s response to SARS-CoV-2 infection might be more influenced by intrinsic individual factors or the overall viral load, rather than the genomic specificity of the viral subvariant.

### 3.8. Cytokine Expression Profiles According to Omicron Subvariants, Inflammatory Network, and Symptom Burden

To investigate potential differences in the inflammatory response induced by distinct Omicron subvariants, RT-qPCR-positive individuals were grouped into two clusters based on previously described genomic and epidemiological characteristics. Cluster 1 comprised early Omicron subvariants from the BA.1 lineage (BA.1, BA.1.1, BA.1.14.1, BA.1.1.14, and BA.1.17), while Cluster 2 included more recent subvariants carrying additional spike protein mutations associated with increased immune escape (BA.4, BA.4.1, BA.5, BA.5.1, BA.5.2.1, and BA.5.1.22). Plasma cytokine levels of IL-6 (*p* = 0.006), IL-2 (*p* = 0.026), IL-4 (*p* < 0.001), and IL-10 (*p* = 0.006) were significantly higher in Cluster 1. In contrast, no significant differences were observed for IL-17A, IFN-γ, or TNF. These findings are presented in Figure 4A.

To further characterize the inflammatory profile in COVID-19 patients, we conducted a Spearman correlation analysis to examine the relationship between the plasma concentrations of all measured cytokines. This analysis revealed strong and statistically significant positive correlations, particularly among cytokines linked to Th1 and Th2 responses. Notably, IL-10 showed moderate-to-strong correlations with TNF (ρ = 0.74), IL-6 (ρ = 0.70), IL-4 (ρ = 0.64), and IL-2 (ρ = 0.76). TNF was also strongly correlated with IL-2 (ρ = 0.77) and IL-4 (ρ = 0.57) and moderately with IL-6 (ρ = 0.50). IL-17A showed weaker, yet significant, correlations with IFN-γ (ρ = 0.43) and IL-10 (ρ = 0.43), while IFN-γ had limited associations with other cytokines. These results suggest a coordinated activation of Th1, Th2, and Th17 pathways during acute SARS-CoV-2 infection, pointing to an integrated inflammatory response. These interrelationships are illustrated in Figure 4B.

Finally, to explore whether the magnitude of clinical manifestations could influence cytokine expression, patients were stratified based on symptom burden into two groups: those reporting fewer than eight symptoms and those reporting eight or more. The comparison of cytokine levels between these groups revealed no statistically significant differences for any of the evaluated cytokines (IL-17A, TNF, IFN-γ, IL-10, IL-6, IL-4, IL-2), indicating that systemic cytokine concentrations were not directly modulated by the total number of symptoms experienced. These findings are shown in Figure 4C–I.

### 3.9. Assessment of the Influence of Comorbidities on Plasma Cytokine Concentrations

To assess whether the presence of comorbidities influenced plasma cytokine concentrations, SARS-CoV-2-positive individuals were stratified into two groups: with comorbidities (*n* = 16) and without comorbidities (*n* = 24). Data normality was assessed using the Shapiro–Wilk test, which indicated that most cytokines (IL-17A, IFN-γ, TNF, IL-6, and IL-2) exhibited non-normal distributions (*p* < 0.05), whereas IL-4 (*p* = 0.120) and IL-10 (*p* = 0.011) were treated as normally distributed for analytical purposes.

Accordingly, non-parametric comparisons were performed using the Mann–Whitney U test for IL-17A (U = 182, *p* = 0.614), IFN-γ (U = 180, *p* = 0.370), TNF (U = 164, *p* = 0.784), IL-6 (U = 154, *p* = 0.856), and IL-2 (U = 173, *p* = 0.705). Parametric comparisons using the unpaired Student’s *t*-test were applied to IL-10 and IL-4, which also revealed no statistically significant differences between the groups (IL-10: t = 0.862, *p* = 0.467; IL-4: t = 0.722, *p* = 0.237).

Taken together, these findings suggest that the presence of comorbidities did not significantly influence the plasma levels of any of the evaluated cytokines during acute SARS-CoV-2 infection in this study cohort (Table 2).

### 3.10. Assessment of the Influence of the Number of Doses of the COVID-19 Vaccine on the Frequency of Reported Symptoms

To investigate whether the number of COVID-19 vaccine doses received was associated with the burden of symptoms reported during acute infection, participants were categorized according to the number of doses received (2 or 3) and stratified by the total number of reported symptoms (<8 or ≥8), based on the median symptom count in the cohort.

A Chi-Square test of independence revealed a statistically significant association between vaccination status and symptom frequency (χ^2^ = 3.88; df = 1; *p* = 0.049), as presented in Figure 5. Among individuals who received three doses, a higher proportion (73.7%) reported fewer than eight symptoms, whereas only 26.3% reported eight or more symptoms. In contrast, individuals who received only two doses showed the opposite pattern, with 57.1% reporting eight or more symptoms and 42.9% reporting fewer symptoms.

The odds ratio was calculated at 0.268 (95% CI: 0.0703–1.02), suggesting that individuals who received three doses of the vaccine were less likely to report a higher symptom burden compared to those who received two doses. Although the confidence interval marginally includes 1.0, the result indicates a potential protective effect of the third vaccine dose in mitigating symptom severity during acute SARS-CoV-2 infection.

## 4. Discussion

The clinical and immunological characterization of this cohort of adult individuals from the northern region of Brazil provided a comprehensive evaluation of cytokine expression patterns and their potential correlations with clinical manifestations, viral subvariants, and relevant epidemiological factors during the acute phase of COVID-19 caused by Omicron subvariants.

Compared to the aggressive and characteristic clinical profile of COVID-19 observed during the first pandemic waves, infections caused by the Omicron variant presented a distinct symptom pattern in our cohort. Symptoms such as loss of smell and taste, previously considered hallmarks of SARS-CoV-2 infection, were reported at much lower frequencies, consistent with previously published studies on Omicron waves. This change reflects the reduction in clinical severity described for Omicron, although vulnerable groups remain at risk of severe outcomes [19].

Our analysis also demonstrated that despite the specific symptom overlap between SARS-CoV-2-positive and -negative individuals, the positive group reported a higher overall symptom burden. This reinforces the notion that, although disease severity has decreased compared to previous variants, as discussed elsewhere, Omicron infections still present a broad clinical spectrum that distinguishes them from other respiratory infections [20,21]. Furthermore, the lack of specificity in clinical presentation highlights the essential role of molecular laboratory diagnosis for the accurate identification of COVID-19.

Despite the high frequency of certain symptoms among infected individuals, no statistically significant associations were observed between these symptoms and the identified Omicron subvariants. This suggests that genomic differences among subvariants may not translate into clearly distinguishable clinical phenotypes [22]. This observation could be attributed to functional conservation in viral entry or replication mechanisms across subvariants or, alternatively, to host-related factors, such as immune status, vaccination history, or comorbidities, playing a more decisive role in shaping the clinical outcome [23,24]. Additionally, the absence of statistical associations may also reflect the limited sample size and fragmentation of frequencies across subvariants. Nonetheless, the relatively homogeneous distribution of dominant symptoms across genetically distinct variants supports the hypothesis of a stable clinical profile within the Omicron lineage, despite its considerable evolutionary diversity [25,26].

Importantly, the immunological analysis comparing SARS-CoV-2-positive and -negative individuals demonstrated significantly elevated levels of pro-inflammatory cytokines in the infected group, particularly IFN-γ, TNF, IL-10, IL-2, and IL-4. These results are in agreement with previous studies that reported Th1-dominant activation and variable engagement of Th2 and Th17 pathways during SARS-CoV-2 infection [27,28]. The absence of statistically significant differences in IL-6 and IL-17A may suggest a more controlled inflammatory response in Omicron infections compared to earlier variants, consistent with the clinical observation of reduced severity during Omicron-dominant waves [29,30].

Interestingly, the correlation analysis between specific symptoms and cytokine levels did not yield strong associations. While an initial link was observed between altered taste and IFN-γ, this finding did not remain significant after multiple comparison correction. These findings suggest that within the context of Omicron, clinical symptomatology may not be tightly linked to discrete cytokine levels but rather result from complex immunological interactions shaped by prior immunity, viral load, and individual variability [31,32,33].

The analysis of a potential association between plasma cytokine levels and the identified SARS-CoV-2 subvariants represented a key step in understanding the immunological dynamics induced by infection. In this study, individuals who tested positive for SARS-CoV-2 were stratified according to the Omicron subvariants identified through viral genotyping, with the aim of exploring whether genomic differences among these subvariants could influence the systemic inflammatory profile. The exclusive predominance of Omicron subvariants in our cohort is explained by the temporal window of sample collection, which took place between late 2021 and early 2022—a period during which Omicron had become the globally dominant variant, effectively replacing previous variants of concern, such as Delta [24,34,35].

Despite the genomic diversity observed among these subvariants, the results indicated no statistically significant differences in the plasma concentrations of the cytokines analyzed. These findings suggest that in the clinical context of mild to moderate acute infections caused by Omicron, the systemic pro-inflammatory immune profile remains relatively conserved across circulating subvariants. This supports the hypothesis of a more uniform immune response within this lineage’s evolutionary trajectory [12,36]. This homogeneity may also reflect the strong selective pressure shaping Omicron’s evolution, favoring subvariants with similar host–virus interaction dynamics despite accumulating mutations in the spike protein [10,14].

Of particular note was the comparative analysis of cytokine levels between individuals infected with older (BA.1-lineage) and more recent (BA.4/BA.5-lineage) subvariants. This revealed significantly higher levels of IL-2, IL-4, IL-6, and IL-10 among those infected with earlier Omicron subvariants. These differences may reflect a more robust immune activation during early Omicron waves, potentially due to lower population-level immunity or a different set of spike mutations [25,37]. Conversely, BA.4 and BA.5 are known to possess enhanced immune escape capacity, which may attenuate cytokine induction or modulate the inflammatory kinetics [38,39].

The cytokine correlation matrix revealed a robust positive association between IL-10, TNF, IL-2, and IL-6, indicating a potentially synchronized activation of both pro-inflammatory and immunomodulatory pathways during acute SARS-CoV-2 infection. This pattern suggests that, rather than an uncontrolled inflammatory surge, the immune response in these individuals—infected predominantly by Omicron subvariants—was characterized by a dynamic interplay between effector and regulatory cytokines [40,41,42].

IL-6 and TNF are well-established mediators of acute inflammation, typically upregulated in response to viral infection and often associated with disease severity in COVID-19. Their concurrent positive correlation with IL-10, a key anti-inflammatory cytokine involved in dampening excessive immune activation and preserving tissue integrity, may reflect an active immunoregulatory mechanism aimed at containing local and systemic damage. Similarly, the observed alignment with IL-2, a cytokine pivotal for T cell proliferation and immune homeostasis, reinforces the notion of a coordinated immune response rather than a dysregulated cytokine storm [43,44,45,46].

Such a balanced inflammatory environment is particularly relevant in the context of Omicron-dominant infections, which have been epidemiologically associated with milder clinical presentations compared to earlier variants such as Delta. This coordinated cytokine expression may underlie the relative clinical stability observed in most individuals during this period, offering immunological insight into the mechanisms that contributed to reduced hospitalization rates despite high transmission levels [47,48,49].

Additional analyses of host-related variables demonstrated that the presence of comorbidities was not significantly associated with altered cytokine levels in this cohort. Meanwhile, the number of COVID-19 vaccine doses received showed a significant inverse association with the frequency of reported symptoms, suggesting that booster doses may help attenuate not only disease severity but also overall symptom burden, an effect supported by the recent literature [50,51,52,53].

Together, these findings provide a detailed view of how different Omicron subvariants interact with the host immune system in real-world settings. Despite the absence of a strong association between variant identity and clinical or immunological profiles, this study highlights the importance of monitoring cytokine responses and symptom expression across different waves of infection.

The relevance of these data extends to current and emerging variants such as XBB.1.5, EG.5, and JN.1, which are direct descendants of Omicron. While these newer variants exhibit enhanced immune evasion and transmissibility, they likely maintain conserved patterns of immune activation, particularly within the Th1, Th2, and Th17 axes. Understanding the cytokine dynamics elicited by early Omicron subvariants, as presented here, provides a useful framework for interpreting immunopathological responses to future variants [54,55,56,57,58].

Lastly, although the present study provides valuable insights, several limitations must be acknowledged. These include the relatively small sample size, which is particularly restrictive in subgroup analyses by Omicron subvariant, thereby limiting the statistical power to detect subtle differences; the cross-sectional design, which does not capture temporal changes in cytokine profiles; and the absence of functional immunological assays, which could have provided mechanistic insights into the observed cytokine patterns. Moreover, this study did not perform viral coinfection screening in RT-qPCR-negative individuals, which might have further clarified differential immune activation profiles. It is, therefore, plausible that some participants classified as SARS-CoV-2-negative were, in fact, experiencing other unconfirmed viral infections, which may partially explain the clinical manifestations and cytokine responses observed in this group. Despite these limitations, this study contributes important data on the immunological and clinical characterization of SARS-CoV-2 Omicron subvariants in a Brazilian Amazonian population—a region underrepresented in global COVID-19 research. These findings may help inform public health decisions and provide a foundation for future studies seeking to understand immune responses to emerging variants in diverse demographic and immunogenetic backgrounds.

## 5. Conclusions

This study presents a characterized panel of cytokines related to Th1, Th2, and Th17 pathways, along with the clinical profile of SARS-CoV-2 Omicron subvariant infections. Our findings highlight a high burden of upper respiratory symptoms, consistent with the clinical profile of Omicron-dominant waves, and demonstrate elevated concentrations of Th1, Th2, and Th17-associated cytokines in infected individuals compared to negative individuals. Despite the genetic diversity among subvariants, symptom frequency and cytokine levels were not significantly associated with specific lineages, suggesting a conserved clinical and immunological response pattern within the Omicron lineage.

Additionally, a comparative analysis of cytokine profiles between individuals infected with early versus later Omicron subvariants revealed distinct inflammatory patterns, reinforcing the importance of monitoring viral evolution and its immunological implications.

Together, these findings contribute to a growing body of evidence regarding the immune response to Omicron and provide a valuable reference for interpreting responses to currently circulating and future SARS-CoV-2 variants. They underscore the need for ongoing genomic and immunological surveillance, particularly in underrepresented populations, to guide public health strategies and support targeted actions such as monitoring symptom burden in vulnerable groups and optimizing vaccine booster policies.

## Figures and Tables

**Figure 1 biomedicines-13-02128-f001:**
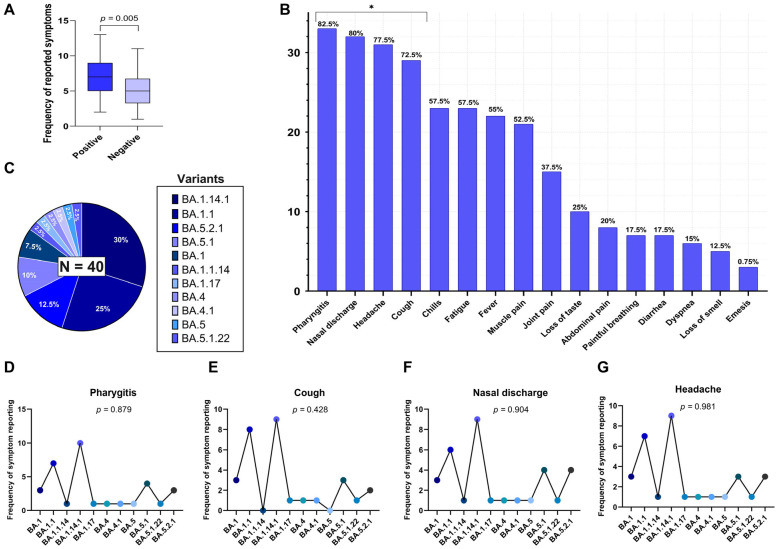
(**A**) Difference in the mean number of reported symptoms between RT-qPCR-positive and –negative individuals for SARS-CoV-2 (Mann–Whitney U test, *p* = 0.005). (**B**) Frequency of reports for 16 investigated symptoms among the positive individuals. Symptoms marked with an asterisk (*) showed a statistically significant distribution (*p* < 0.05), being more frequently reported in the sample, based on a Chi-Square test. (**C**) Distribution of Omicron subvariants identified by genomic sequencing among the SARS-CoV-2-positive individuals (*n* = 40). (**D**–**G**) Frequency of the most frequently reported symptoms (pharyngitis, cough, nasal discharge, and headache) distributed across subvariants, with corresponding *p*-values from Fisher’s exact test, indicating no significant association between subvariants and symptom occurrence.

**Figure 2 biomedicines-13-02128-f002:**
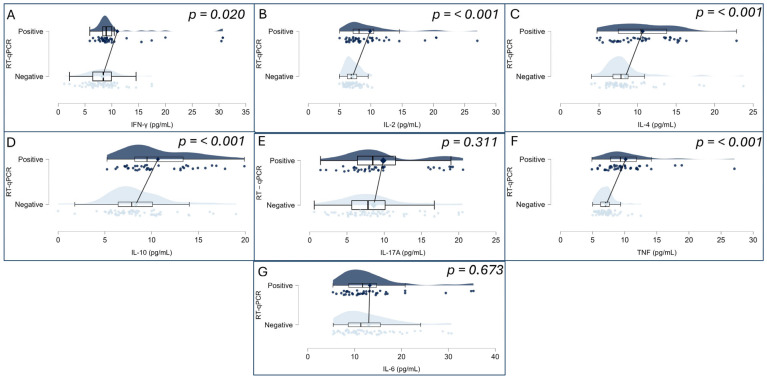
Cytokine levels (pg/mL) were measured for IFN-γ (**A**), IL-2 (**B**), IL-4 (**C**), IL-10 (**D**), IL-17A (**E**), TNF (**F**), and IL-6 (**G**). Each panel combines violin plots (representing data distribution density), boxplots (indicating median, interquartile range, and range), and individual data points (jittered for visibility). Groups are divided by RT-qPCR result: SARS-CoV-2-positive (**top**, dark blue) and SARS-CoV-2-negative (**bottom**, light blue). Statistical comparisons were performed using the non-parametric Mann–Whitney U test (*p*-values displayed on each panel). Statistically significant differences (*p* < 0.05) were observed for IFN-γ, TNF, IL-2, IL-4, and IL-10.

**Figure 3 biomedicines-13-02128-f003:**
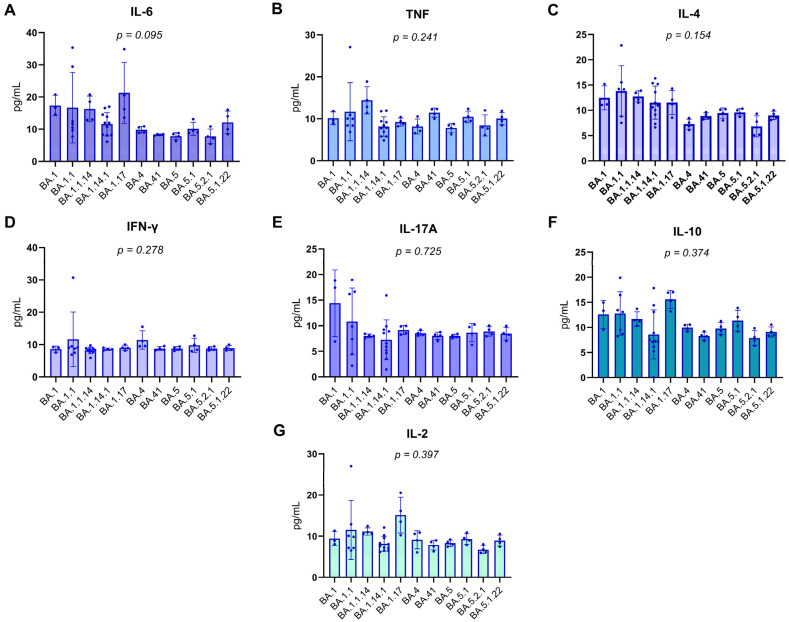
Distribution of Plasma cytokine levels in RT-qPCR-positive individuals. Boxplots represent the concentration (pg/mL) of IL-17A, IFN-γ, TNF, IL-10, IL-6, IL-4, and IL-2 across different Omicron subvariants. The central lines indicate the median; boxes represent the interquartile range (IQR), and whiskers denote minimum and maximum values. Shapiro–Wilk test *p*-values are shown above each cytokine, indicating deviations from normality in most distributions (*p* < 0.05), except for IL-10 and IL-4, which did not significantly depart from normality. Notably, the variants on the *X*-axis are arranged in chronological order, from the earliest to the most recently emerged subvariant (left to right), enabling visual analysis of temporal patterns in cytokine expression. Associations between cytokine levels and viral subvariants were statistically evaluated using the Kruskal–Wallis test, appropriate for non-parametric comparisons among multiple independent groups.

**Figure 4 biomedicines-13-02128-f004:**
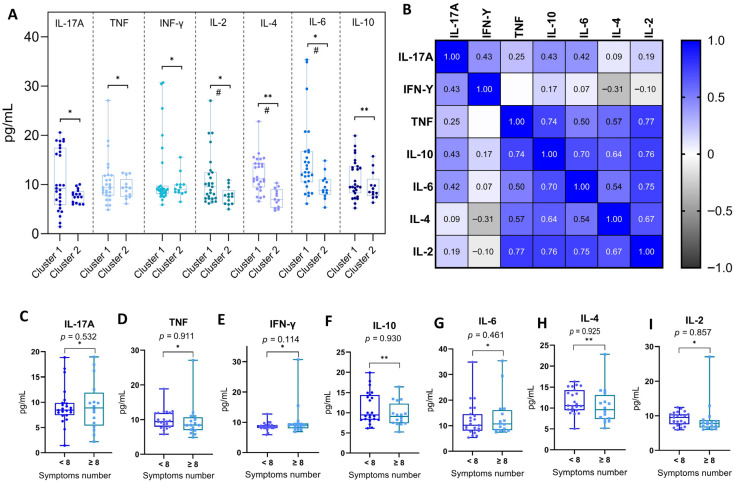
Integrated analysis of plasma cytokine concentrations and clinical correlates in individuals positive for SARS-CoV-2. (**A**) Comparison of plasma cytokine concentrations between individuals infected with early (Cluster 1) and late (Cluster 2) Omicron subvariants. Cluster 1 comprises individuals infected with Omicron BA.1-lineage subvariants (BA.1, BA.1.1, BA.1.14.1, BA.1.1.14, and BA.1.17), while Cluster 2 includes subvariants from the BA.4/BA.5 lineage (BA.4, BA.4.1, BA.5, BA.5.1, BA.5.2.1, and BA.5.1.22). Cytokines evaluated were as follows: IL-17A, TNF, IFN-γ, IL-2, IL-4, IL-6, and IL-10. Statistical comparisons were performed using the Mann–Whitney U test (*) for non-normal distributions and the Student’s *t*-test (**) for normal distributions, as determined by the Shapiro–Wilk test. Significant differences (*p* < 0.05) are marked with (#). (**B**) Spearman correlation matrix of plasma cytokine concentrations in SARS-CoV-2-positive individuals. The heatmap displays pairwise correlation coefficients between IL-17A, IFN-γ, TNF, IL-10, IL-6, IL-4, and IL-2. Color intensity reflects the strength and direction of correlation, from negative (black) to positive (blue). Only statistically significant correlations (*p* < 0.05) are presented. (**C**–**I**) Comparison of plasma cytokine concentrations in SARS-CoV-2-positive individuals according to symptom burden. Patients were stratified based on the total number of reported symptoms during acute infection: fewer than 8 symptoms (<8) or 8 or more symptoms (≥8), based on the cohort median. Cytokines analyzed include IL-17A (**C**), TNF (**D**), IFN-γ (**E**), IL-10 (**F**), IL-6 (**G**), IL-4 (**H**), and IL-2 (**I**). Boxplots show the median, interquartile range (IQR), and full data range. Statistical comparisons were made using the Mann–Whitney U test (*) for non-normal distributions and the Student’s *t*-test (**) for normal distributions. No statistically significant differences were observed between groups (*p* > 0.05 for all comparisons).

**Figure 5 biomedicines-13-02128-f005:**
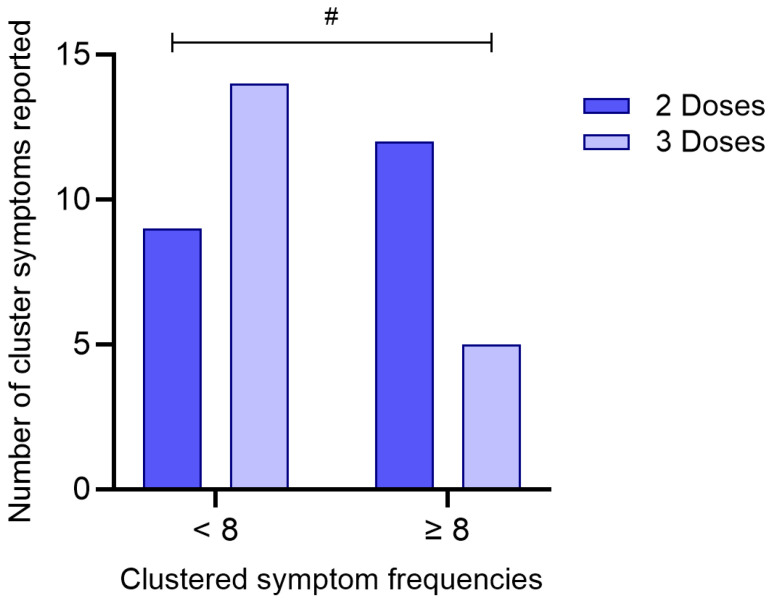
Association between the number of COVID-19 vaccine doses received and clustered symptom frequencies. Participants were grouped based on the number of vaccine doses received (2 or 3) and stratified according to the total number of reported symptoms (<8 or ≥8). The bar graph represents the absolute number of individuals in each category. A statistically significant difference between groups was identified using the Chi-Square test (*p* = 0.049). (#) denotes *p* < 0.05.

**Table 1 biomedicines-13-02128-t001:** Sociodemographic, Clinical, and Symptom Characteristics of the Study Population According to SARS-CoV-2 RT-qPCR results.

Characteristics (*n* = 115)	RT-qPCR		*p*-Value *
	Positive *n* = 40 (%)	Negative *n* = 75 (%)	
Age Range (years)			0.707
<25	2 (5.0)	7 (9.3)	
25–50	30 (75.0)	53 (70.7)	
>50	8 (20.0)	15 (20.0)	
Sex			0.624
Male	9 (22.5)	20 (26.7)	
Female	31 (77.5)	55 (73.3)	
Comorbidity			0.029
Yes	16 (40.0)	21 (28.0)	
No	24 (60.0)	54 (72.0)	
Symptoms			
Pharyngitis	33 (82.5)	36 (48.0)	0.136
Cough	29 (72.5)	11 (14.7)	0.626
Fever	22 (55.0)	28 (37.3)	0.626
Nasal discharge	32 (80.0)	42 (56.0)	0.528
Fatigue	23 (57.5)	27 (36.0)	0.528
Headache	31 (77.5)	40 (53.3)	0.528
Joint pain	15 (37.5)	12 (16.0)	0.212
Muscle pain	21 (52.5)	29 (38.6)	0.840
Painful breathing	7 (17.5)	3 (4.0)	0.212
Dyspnea	6 (15.0)	10 (13.3)	0.913
Diarrhea	7 (17.5)	10 (13.3)	0.913
Chills	23 (57.5)	12 (16.0)	0.014
Abdominal pain	8 (20.0)	9 (12.0)	0.685
Loss of taste	10 (25.0)	9 (12.0)	0.528
Loss of smell	5 (12.5)	7 (9.33)	0.913
Emesis	3 (7.5)	8 (10.6)	0.626

*n* = number of individuals. * Chi-Square test.

**Table 2 biomedicines-13-02128-t002:** Comparison of Plasma Cytokine Levels in SARS-CoV-2-Positive Individuals With and Without Comorbidities.

Cytokine	Comorbidity (m *)		*p*-Value
	Yes *n* = 16	No *n* = 24	
IL-17A	9.75	9.95	0.614 ^a^
TNF	9.85	10.34	0.784 ^a^
IFN-y	10.02	11.93	0.370 ^a^
IL-10	10.61	10.7	0.467 ^b^
IL6	14.00	12.71	0.856 ^a^
IL-4	9.99	10.9	0.237 ^b^
IL-2	9.35	10.23	0.705 ^a^

*n* = number of individuals; m * mean cytokine dosage in pg/mL, ^a^ Mann–Whitney U test; ^b^ Student’s *t*-test.

## Data Availability

The authors declare that this research was conducted in the absence of any commercial or financial relationships that could be construed as potential conflicts of interest.

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
