# Peer review of "Clinical Manifestations and Cytokine Profiles of the Th1, Th2, and Th17 Response Associated with SARS-CoV-2 Omicron Subvariants"

_biomedicines, 2025, doi:10.3390/biomedicines13092128_

Round 1

Reviewer 1 Report

Comments and Suggestions for Authors

Your manuscript provides a valuable and well-presented analysis of the clinical and immunological characteristics of infections caused by SARS-CoV-2 Omicron subvariants. The integration of genomic, clinical, and cytokine data is a significant strength, and your findings, particularly the comparison between early and later Omicron clusters and the data on vaccine effectiveness, are important contributions to the field. The paper is clearly written and logically structured. I recommend it for publication after you address the following points to improve its clarity and accuracy:

  • In section 3.2, there is a minor inconsistency in the p-value for the total number of reported symptoms. The text states p=0.005, while the caption for Figure 1A states p=0.0096. Please correct this for consistency.
  • In section 3.7, you mention that an association did not remain significant after "adjustment for multiple comparisons". Please specify the correction method you applied (e.g., Bonferroni, FDR) and provide the adjusted p-value.
  • In section 3.12, you state that IL-10 (p=0.011) was treated as normally distributed following a Shapiro-Wilk test. A p-value below 0.05 typically indicates a non-normal distribution. Please clarify this choice or re-analyze with a non-parametric test for consistency.
  • The manuscript alternates between "plasma" and "serum." Your methods section indicates that plasma was collected and used for cytokine quantification, but terms like "serum concentrations" and "serum cytokine levels" are used later. Please standardize the terminology throughout the manuscript for accuracy.
  • Your funding statement declares "no external funding", but the acknowledgments thank the "National Council for Scientific and Technological Development (CNPq)". Please clarify this. If any support (such as grants or fellowships) was provided by CNPq, it should be formally listed in the funding section.

Author Response

Comments 1: "In section 3.2, there is a minor inconsistency in the p-value for the total number of reported symptoms. The text states p = 0.005, while the caption for Figure 1A states p = 0.0096. Please correct this for consistency".

Response 1:
Thank you for pointing this out. We agree with the reviewer’s observation regarding the inconsistency in the p-value. Therefore, we have revised the figure caption to ensure consistency with the results described in the text. The corrected p-value is p = 0.005.

This change can be found in the revised manuscript on page 6, Figure 1 caption:

Figure 1. (A) Difference in the mean number of reported symptoms between RT-qPCR–positive and –negative individuals for SARS-CoV-2 (Mann-Whitney U test, p = 0.005).

We have ensured that both the main text and figure caption now report the same p-value for clarity and accuracy.

Comment 2: "Please specify the correction method you applied (e.g., Bonferroni, FDR) and provide the adjusted p-value."

Response 2:
Thank you for your valuable observation. We agree with your comment and have updated the manuscript accordingly. To account for multiple comparisons, we applied the Benjamini-Hochberg correction method. Although the nominal p-value for the association between loss of taste and IFN-γ levels was 0.038, this result did not remain significant after correction (adjusted p = 0.955). This clarification has now been included in the manuscript (page 8, lines 280-281).
[Updated text in the manuscript:]
"Only one comparison yielded a nominal p-value below the conventional significance threshold: the association between loss of taste and IFN-γ levels (p = 0.038). However, this result did not remain significant after correction for multiple comparisons using the Benjamini-Hochberg method (adjusted p = 0.955)."

Comment 3:
In section 3.12, you state that IL-10 (p=0.011) was treated as normally distributed following a Shapiro-Wilk test. A p-value below 0.05 typically indicates a non-normal distribution. Please clarify this choice or re-analyze with a non-parametric test for consistency.

Response 3:
Thank you for your valuable observation. You are absolutely correct in highlighting this inconsistency. Upon reviewing the manuscript, we identified that the p-value for IL-10 was erroneously reported as 0.011. The correct p-value obtained from the Shapiro–Wilk normality test is p = 0.258, which supports the assumption of a normal distribution. This justifies the application of the unpaired Student’s t-test for this cytokine. We have corrected the text accordingly in the revised manuscript (Page 12, Lines 373–374).

[Updated text in the manuscript: “[…] whereas IL-4 (p = 0.120) and IL-10 (p = 0.258) were treated as normally distributed for analytical purposes.”]

Comment 4:
The manuscript alternates between "plasma" and "serum." Your methods section indicates that plasma was collected and used for cytokine quantification, but terms like "serum concentrations" and "serum cytokine levels" are used later. Please standardize the terminology throughout the manuscript for accuracy.

Response 4:
Thank you for your careful observation. We agree with your comment and acknowledge the importance of consistent terminology. As plasma samples were indeed used for cytokine quantification in our study, we have revised the manuscript to standardize the use of the term “plasma” throughout the entire text for clarity and accuracy.

Comment 5:
Your funding statement declares "no external funding", but the acknowledgments thank the "National Council for Scientific and Technological Development (CNPq)". Please clarify this. If any support (such as grants or fellowships) was provided by CNPq, it should be formally listed in the funding section.

Response 5:
Thank you for this important observation. We agree with the comment and clarify that the study itself did not receive direct funding for its execution. However, one of the authors (M.A.B.) was awarded a research support fellowship by the National Council for Scientific and Technological Development (CNPq), which helped cover general academic expenses. To address this and provide transparency, we have updated the "Funding" section of the manuscript accordingly. The revised text now reads:
“Funding: This study did not receive specific funding from public, commercial, or non-profit funding agencies. However, M.A.B. received a research support scholarship from the National Council for Scientific and Technological Development (CNPq), under grant number #178654/2024-8.”
This update can be found on page 16, lines 551 to 553 of the revised manuscript.

A .DOCX file has been attached to this response with all changes made to the manuscript for better transparency in the submission process.

Reviewer 2 Report

Comments and Suggestions for Authors

Cytokine profiles of the Th1, Th2 and Th17 response and Clinical Manifestations Associated with SARS-CoV-2 Omicron Subvariants

There are however a few observations that may require clarification:

MAIN COMMENTS:

This is an interesting study and it would be helpful for the reader to consider the comparison of negative and Positive SARS-CoV-2 RT-PCR individual?

Was this a comparison of the SARS-CoV-2 variants?

How does this information impact the other variants?

is this information novel or is likely in other variants?

Do the symptoms or the combinations really matter among the variants?

Is this information applicable to other variants? 

Is it not expected that the difference would obviously be there?

Did the author consider comparing responses among positive individuals rather that comparing the infected and the non-infected?  The methods do not clearly explain to allow appreciation of the methodology used.

it would be helpful if the authors could reconsider selecting a few pertinent aspects of the write and converge a narrow objective. The Methods would also need to be expanded to justify the contents of the results such that the reader would have a good guess of the expected results. 

I would suggest that the results be summarized and many of the headed omitted and the others merged into a more data friendly package because making a comparison of every possible analysis dilutes the message delivered by the write up.

Title:

The title may need to be reconsidered because the Th1, Th2 and Th17 appears after the clinical manifestation. I suggest that the clinical manifestation is presented first. In addition, this is the natural order of occurrence of the two parts. Clinical manifestation, then cytokine profiles.

Abstract

Line 34: refers to the present” has become” is this in relation to the study period or the current COVID-19 status considering that the Pandemic has since ended? Please clarify. Please review section 2.1; Line 98-101.

Methods

General comment: The methods are not clear enough to allow the reader to understand how the study was conducted except that there were 115 individuals. i suggest that the methods be elaborately and concisely written to enable appreciation of the study methodology and reasoning or sufficient citation be provided.

Line 112: requires a citation;

Line 114-130: was this section performed in this study or earlier. If so, there is need for a citation to explain clearly what was done e.g., this citation would negate the use of words such as “specialized bioinformatics software” (c.f. Line 123).

Line 132-143: This section requires a detailed explanation similar to the comment about unlike using “distinct set of beads” as this doesn’t help the reader to understand what was done.

Line 145-166: Please consider a non-academic approach to writing this section by avoiding using the terms such as continuous and categorical variables. Instead, I would suggest using the actual scientific basis of the statistical analysis. e.g., Spearman’s rank correlation was used to explore the association of the number of symptoms and cytokines levels xxxxxxxx. As such all the continuous and categorical or numerical variables maybe omitted as it is assumed that the reader may understand the type of variables they could be.

Line 154: Consider using the accepted format for Chi-Squared test, with a capital “C” and “Squared”

Results

General comment:

There is an observation that there is single page of Materials and Methods compared to 10.5 pages of results. This difference is quite disproportionate. The appearance may suggest the following:

  1. The methods are incomplete
  2. There was an over analysis of the results
  • The results not be concise and not summarized adequately.

May I suggest that the 13 subsections be collapsed in a handful section to explain the purpose of the study.

3.1. Demographic characterization and diagnostic confirmation

3.2. Symptomatologic comparison between groups

3.3. Symptoms presented by the positive group

3.4. Omicron subvariants identified in the sample

3.5. Correlation of the most frequent symptoms of the positive group with the identified variants

3.6. Comparison of Plasma Cytokine Levels Between SARS-CoV-2 Positive and Negative Individuals

3.7. Association between clinical symptoms and serum cytokine levels in patients with acute COVID-293 19

3.8. Correlation between cytokine levels and identified Omicron subvariants

3.9. Comparison of Serum Cytokine Levels Between Omicron Subvariant Clusters

3.10. Interrelationships Among Serum Cytokine Levels in Infected Individuals

3.11. Correlation Between Symptom Burden Clusters and Plasma Cytokine Levels

3.12. Assessment of the influence of comorbidities on plasma cytokine concentrations

3.13. Assessment of the influence of the number of doses of the COVID-19 vaccine on the frequency of 440 reported symptoms

Line 180-190: consider deleting or summarizing this section a single sentence without losing much information.

Line 193-194: The statement regarding the “reported at least one comorbidity” is confusing when using the absolute number; c.f. 16 vs 21. I suggest using the percentage unless the denominator is included in the statement, or the absolute values be preceded by the percentage e.g., 40% (21/xxx) compared to 28% (21/XX).

Conclusion

The conclusion does not address the core of the study such that the vaccine effect is appearing when this was not adequately explained in the whole write up.

Author Response

Comment 1: "This is an interesting study and it would be helpful for the reader to consider the comparison of negative and Positive SARS-CoV-2 RT-PCR individual?"

Response 1: Thank you for your thoughtful comment and for recognizing the relevance of our study. We agree that comparing SARS-CoV-2 positive and negative individuals adds important context and strengthens the interpretation of our findings. In this study, we included such comparisons to highlight how individuals with confirmed COVID-19 differ from those presenting with similar respiratory symptoms but with negative RT-qPCR results. Specifically, we compared the distribution and frequency of self-reported symptoms, as well as cytokine concentrations, between these two groups. This approach aimed to underscore the distinct clinical and immunological features associated with SARS-CoV-2 infection, even within a population where other respiratory pathogens may be circulating. We believe that this comparison provides a clearer understanding of the immunological patterns linked specifically to COVID-19 and enhances the overall robustness of our conclusions.

Comment 2: "Was this a comparison of the SARS-CoV-2 variants?"

Response 2: Thank you for your question. Yes, one of the objectives of our study was to compare different subvariants of the SARS-CoV-2 Omicron lineage. We aimed to evaluate whether infections caused by distinct Omicron subvariants were associated with differences in clinical manifestations and cytokines immune profiles. To this end, we analyzed the expression levels of key cytokines involved in Th1-, Th2-, and Th17-type immune responses, as well as symptom frequency, seeking to understand potential immunological and clinical distinctions among subvariants.

Comment 3:
"How does this information impact the other variants?"

Response 3:
Thank you for this thoughtful question. While our study specifically focused on Omicron subvariants circulating in our region, the findings may have broader implications for the understanding of immune responses to other SARS-CoV-2 variants. The relative preservation of cytokine expression patterns, especially involving Th1, Th2, and Th17 pathways, despite genetic divergence among Omicron subvariants, suggests a certain degree of immunological consistency in the host response. This insight could serve as a comparative reference point for assessing how immune responses to earlier or future variants differ or remain conserved. By establishing this immunological baseline for Omicron, our results may help inform future evaluations of emerging lineages in terms of their pathogenicity and capacity to modulate immune activation.

Comment 4:
"Is this information novel or is likely in other variants?"
Response 4: Thank you for raising this thoughtful question. While some cytokine response patterns, particularly involving Th1-associated cytokines like IFN-γ and IL-2, have been described in infections caused by earlier SARS-CoV-2 variants, our study adds novel data by also evaluating the Th2 (e.g., IL-4, IL-10) and Th17 (e.g., IL-17A) immune axes in the context of Omicron subvariants. These arms of the immune response have been less extensively explored in Omicron infections, especially in our region.

Importantly, our findings suggest that although Th1 activation remains predominant, there is also measurable engagement of Th2 and Th17 pathways, highlighting a broader, coordinated immune response. This observation contrasts with earlier variants where a hyperinflammatory Th1-dominated profile was more common, and it may partially explain the more controlled clinical manifestations seen with Omicron.

Moreover, our study offers original insight by characterizing these responses in a population from the Brazilian Amazon, a region with distinct demographic, genetic, and epidemiological features and which remains underrepresented in global COVID-19 immunology research. Thus, while some features of the cytokine response may overlap with previous variants, the comprehensive evaluation of Th1, Th2, and Th17 cytokines in this unique context makes this information both novel and valuable.

Comment 5:
"Do the symptoms or the combinations really matter among the variants?"

Response 5:
Thank you for this insightful question. In our study, we explored the distribution of self-reported symptoms among individuals infected with different Omicron subvariants to assess whether distinct clinical profiles could be identified. Our analysis did not reveal statistically significant associations between specific symptoms (or combinations thereof) and particular subvariants.

This suggests that, within the Omicron lineage, despite its considerable genetic diversification, clinical manifestations remain relatively homogeneous. One possible explanation is that the mutations distinguishing these subvariants may not result in major functional changes in viral tropism or pathogenicity. Alternatively, host, related factors, such as vaccination status, prior exposure, and immunogenetic background, may play a more influential role in shaping symptom profiles than the specific subvariant.

Therefore, although symptoms and their combinations are relevant for clinical diagnosis and public health surveillance, in the context of Omicron subvariants, they may not serve as reliable indicators to distinguish between circulating sublineages. This observation reinforces the importance of molecular tools for accurate viral identification and variant tracking.

Comment 6:
Is this information applicable to other variants?

Response 6:
Thank you for your thoughtful question. While our study focused specifically on individuals infected with Omicron subvariants, several findings may indeed have broader relevance to other SARS-CoV-2 variants. For example, the coordinated activation of Th1-, Th2-, and Th17-associated cytokines, particularly the interplay between IFN-γ, IL-2, IL-4, IL-6, IL-10, and TNF, has been observed in earlier phases of the pandemic and across different variants, suggesting that certain immunological mechanisms are conserved in SARS-CoV-2 infection regardless of the variant.

However, we also acknowledge that each variant may interact uniquely with host immunity due to differences in transmissibility, immune escape capacity, and spike protein mutations. Thus, although some of the immunological patterns we observed may apply to other variants, particularly those derived from Omicron (e.g., XBB.1.5, EG.5, JN.1), direct extrapolation should be approached with caution.

Our findings can serve as a useful framework for comparative analyses in future studies, especially those evaluating immune responses to emerging variants in populations with similar demographic and immunogenetic backgrounds.

Comment 7:
"Is it not expected that the difference would obviously be there?"

Response 7:
Thank you for raising this important point. While certain immunological differences between infected and non-infected individuals might indeed be expected, such as higher levels of inflammatory cytokines in those testing positive for SARS-CoV-2, we considered it essential to empirically validate these assumptions in our specific population.

Our cohort is composed of individuals from the Brazilian Amazon, a region with distinct genetic, environmental, and epidemiological characteristics that may influence the immune response. Furthermore, given that the Omicron subvariants are known for their immune-evasive properties and often cause milder clinical disease, the magnitude and pattern of cytokine responses in this setting were not fully predictable.

Thus, even findings that may seem intuitive benefit from confirmation through population-specific analyses, especially considering the limited representation of Amazonian populations in global immunological studies of COVID-19. These data provide important context for ongoing and future comparisons involving other variants and vaccination statuses.

Comment 8:
"Did the author consider comparing responses among positive individuals rather than comparing the infected and the non-infected? The methods do not clearly explain to allow appreciation of the methodology used."

Response 8:
We sincerely thank the reviewer for this thoughtful observation. In fact, we did conduct several analyses specifically within the SARS-CoV-2–positive group, in addition to comparisons between infected and non-infected individuals.

Our between-group comparisons (positive vs. negative) were designed to verify whether COVID-19 still presented a distinguishable clinical and immunological profile from other respiratory illnesses in a vaccinated population during the Omicron wave. However, to better explore the heterogeneity within the infected cohort, we carried out multiple internal analyses exclusively among SARS-CoV-2–positive individuals.

These include:

Section 3.2 (p. 6, lines 217–227): Analysis of symptom frequencies and symptom burden within the positive group.

Section 3.3: Identification and description of Omicron subvariants via genotyping.

Section 3.4: Evaluation of associations between subvariants and specific symptom profiles.

Sections 3.6 and 3.7: Investigation of relationships between cytokine levels and clinical symptoms among infected individuals.

Sections 3.8: Comparison of cytokine profiles among subvariants, with temporal clustering (early vs. late subvariants).

Section 3.9: Spearman correlation matrix assessing potential monotonic relationships among cytokines in the positive group.

Section 3.10: Stratification based on symptom burden (above vs. below median) to assess cytokine differences.

Section 3.11: Analysis of the impact of comorbidities on cytokine levels within the infected cohort.

We hope this clarifies that intra-group analyses were indeed a central part of our study. To address the reviewer’s concern, we have also reviewed the Methods section to ensure clearer descriptions of the stratifications and analytical strategies applied specifically to the SARS-CoV-2–positive group.

Comment 9: "it would be helpful if the authors could reconsider selecting a few pertinent aspects of the write and converge a narrow objective. The Methods would also need to be expanded to justify the contents of the results such that the reader would have a good guess of the expected results."

Response 9: We sincerely thank the reviewer for this thoughtful suggestion. We acknowledge the importance of clearly defined objectives and streamlined presentation. However, we respectfully emphasize that each of the analyses presented in our manuscript was carefully planned based on specific scientific questions raised during the study. We believe that the integration of multiple variables, such as symptoms, cytokine profiles, subvariants, vaccination status, and comorbidities, offers valuable insights into the multifactorial nature of SARS-CoV-2 Omicron infections and their immune-clinical presentation. Rather than diluting the focus, we aimed to provide a comprehensive yet coherent dataset that reflects the complexity of the subject. To address the reviewer’s concern, we have revised the final paragraph of the introduction to more clearly describe the scope and objectives of the study, ensuring that the reader can anticipate the rationale for each result presented. We have also reviewed the Methods section and made expansions where necessary to strengthen the clarity and alignment between our methodological design and the analyses presented in the Results section.

Comment 11 – Reviewer 2:
“I would suggest that the results be summarized and many of the headed omitted and the others merged into a more data friendly package because making a comparison of every possible analysis dilutes the message delivered by the write up.”

Response 11:
We thank the reviewer for this valuable observation. We agree with the importance of delivering a more concise and cohesive narrative in the Results section to avoid diluting the overall message. In response, we have revised the structure of the Results section to streamline the presentation of findings. Specifically, we merged and summarized the content from former subsections 3.2 and 3.5 into a unified subsection now titled 3.2. Symptomatologic Analysis, which presents a comprehensive but focused view of the symptomatologic findings.

Additionally, the results from subsections 3.8, 3.9, 3.10, and 3.11 have been condensed and integrated into a single subsection titled 3.8. Cytokine Expression Profiles According to Omicron Subvariants, Inflammatory Network, and Symptom Burden. This restructuring aims to deliver the findings in a more data-friendly and interpretable format, while maintaining scientific rigor and clarity.

We appreciate the reviewer’s suggestion, which we believe has enhanced the readability and impact of our manuscript.

Comment 12:
The title may need to be reconsidered because the Th1, Th2 and Th17 appears after the clinical manifestation. I suggest that the clinical manifestation is presented first. In addition, this is the natural order of occurrence of the two parts. Clinical manifestation, then cytokine profiles.

Response 12:
Thank you for this insightful suggestion. We agree that listing the clinical manifestations before the cytokine profiles follows a more natural and chronological structure. Therefore, we have revised the title of the manuscript to:
"Clinical Manifestations and Cytokine Profiles of the Th1, Th2, and Th17 Response Associated with SARS-CoV-2 Omicron Subvariants."
We believe this change improves the clarity and logical flow of the title, aligning it more closely with the structure of the manuscript and the reader's expectations.

Comments 13:
"Line 34: refers to the present” has become” is this in relation to the study period or the current COVID-19 status considering that the Pandemic has since ended?"

Response 13:
Thank you very much for this important grammatical observation. We agree that clarifying the temporal context is crucial for accurate interpretation. Therefore, we have revised the sentence to better reflect the timeframe of the study and the status of the COVID-19 pandemic. The updated text now reads:
"Background: The SARS-CoV-2 Omicron variant became a dominant driver during the COVID-19 pandemic due to its high transmissibility and immune escape potential."
This change can be found in the Abstract, lines 33-34, page 1.

Comments 14:
"Line 112: requires a citation"

Response 14:
Thank you for highlighting the need for a citation in this section. We acknowledge the importance of providing appropriate references to support methodological details. However, in this case, the sequencing was performed by collaborators who are listed as co-authors and actively contributed to the writing of this methodology section. Therefore, we considered the description as original data generated within our research team.

Comment 15:
“Line 114–130: Was this section performed in this study or earlier? If so, there is need for a citation to explain clearly what was done e.g., this citation would negate the use of words such as 'specialized bioinformatics software' (c.f. Line 123).”

Response 15:
We thank the reviewer for this insightful observation. In response, we revised the corresponding section of our manuscript to provide a clearer and more precise description of the genomic analysis and lineage classification performed in our study. Specifically, we specified the name of the workflow used, ViralFlow, and described its main steps, including quality control, genome assembly, and lineage assignment using PANGOLIN and Nextclade. These procedures are part of the ongoing genomic surveillance efforts for SARS-CoV-2 in Brazil. As part of this initiative, our samples were submitted to GISAID, as detailed in the Methodology. The information on variants used in our analyses was retrieved directly from GISAID.

The revised paragraph now reads as follows:
“The genomic analysis and classification of SARS-CoV-2 lineages and sublineages were performed using the ViralFlow workflow [15,16]. This pipeline allows for automated processing of sequencing data, including quality control, genome assembly, and lineage assignment. Consensus sequences were generated from high-quality reads and classified using the PANGOLIN tool. Clade assignment and sequence quality were further evaluated using the Nextclade platform.”

(Section 2. Results, Topic 2.2 Genotyping Data Acquisition, lines 122-127)

Comment 16:
"Line 132–143: This section requires a detailed explanation similar to the comment about unlike using 'distinct set of beads' as this doesn’t help the reader to understand what was done."

Response 16:
Thank you for your observation. We understand that the phrase “distinct sets of beads” may have caused confusion. To avoid misinterpretation, we have removed this simplified technical description. As the assay followed the manufacturer’s protocol in full detail, we maintained only the reference to the commercial kit and its instructions.

Comment 17:
“Please consider a non-academic approach to writing this section by avoiding using the terms such as continuous and categorical variables. Instead, I would suggest using the actual scientific basis of the statistical analysis. e.g., Spearman’s rank correlation was used to explore the association of the number of symptoms and cytokines levels xxxxxxxx. As such all the continuous and categorical or numerical variables maybe omitted as it is assumed that the reader may understand the type of variables they could be.”

Response:
We sincerely thank the reviewer for this valuable suggestion. We agree that a more scientific and objective description of the statistical methods improves clarity and aligns better with the expectations of a broader readership. Accordingly, we have revised the Statistical Analysis section to omit references to variable types and instead directly describe the statistical tests used and their contexts. The revised section now reads as follows:

All statistical analyses were conducted using JASP (version 0.17.2) and GraphPad Prism (version 10.5.0). Descriptive statistics were first applied to summarize clinical, demographic, and immunological data. The normality of data distributions was assessed using the Shapiro–Wilk test. Group comparisons were performed using Student’s t-test for normally distributed variables. For variables that deviated from normality, the Mann–Whitney U test was used. the Kruskal–Wallis test was applied to be applied to investigate associations between identified subvariants and cytokine levels. Associations between clinical characteristics, such as symptom occurrence and SARS-CoV-2 infection status, or comorbidities, were evaluated using the chi-square test or Fisher’s exact test, depending on the sample size and distribution of responses. Spearman’s rank correlation was used to investigate monotonic relationships between plasma cytokine levels associated with Th1, Th2, and Th17 responses. This analysis was also applied to examine potential associations between symptom burden and cytokine expression. A significance level of p < 0.05 and a 95% confidence interval were adopted throughout.” 

This revised version can be found in the Methods section, item 2.4 “Statistical Analysis”, on page 4, lines 145–158.

Comment 18:
"There is an observation that there is a single page of Materials and Methods compared to 10.5 pages of results. This difference is quite disproportionate. The appearance may suggest the following:

The methods are incomplete

There was an over analysis of the results

The results may not be concise and not summarized adequately.

May I suggest that the 13 subsections be collapsed into a handful of sections to explain the purpose of the study."

Response 18:
We thank the reviewer for this insightful and constructive comment. We understand the concern regarding the proportion between the Materials and Methods and Results sections, as well as the structure of the numerous subsections presented.

To improve the clarity and purpose of our manuscript, we have revised the final paragraph of the Introduction to more clearly outline the study objectives and to provide the reader with a structured preview of what to expect in the following sections. The updated text reads: “Therefore, this study aimed to evaluate whether SARS-CoV-2 Omicron subvariants circulating in northern Brazil were associated with differential expression of a targeted set of cytokines linked to Th1, Th2, and Th17 immune responses. Additionally, we investigated how these cytokine profiles relate to clinical characteristics such as symptom burden, comorbidities, and specific subvariant clusters. By focusing on these key immunological pathways, we sought to describe relevant immuno-clinical associations observed in this regional cohort during the Omicron wave.
(Introduction, pages 2–3, sixth paragraph, lines 91–97).

To improve the clarity and scientific transparency of our manuscript, we have revised the Materials and Methods section to better detail the procedures used, particularly regarding the identification of Omicron subvariants. Specifically, we expanded the description of Subsection 2.2. Genotyping Data Acquisition (second and third paragraphs, lines 122–131), in order to clarify how genotyping data were obtained and integrated into the study.

In response to the recommendation to streamline the Results section, we have restructured it by merging and condensing several subsections to reduce redundancy and improve the overall reading experience. Specifically:

We merged and summarized the content from former subsections 3.2 and 3.5 into a unified subsection titled “3.2. Symptomatologic Analysis”, which presents a more focused overview of symptom-related findings.

Additionally, the results previously described in subsections 3.8, 3.9, 3.10, and 3.11 were consolidated into a single, comprehensive subsection titled “3.8. Cytokine Expression Profiles According to Omicron Subvariants, Inflammatory Network, and Symptom Burden.

These changes aim to enhance the manuscript’s readability, provide a clearer narrative structure, and maintain scientific rigor. We believe the revised version now offers a more concise and coherent presentation of our findings, while still addressing the full scope of our research objectives.

Comments 19:
"Line 180–190: consider deleting or summarizing this section into a single sentence without losing much information."

Response 19:
Thank you for this helpful suggestion. We agree that this section could be more concise without compromising the information conveyed. In response, we revised the text to present the data in a more summarized format while retaining all relevant demographic details. The updated paragraph now reads:
Participants were stratified into three age groups: <25, 25–50, and >50 years. Among SARS-CoV-2–positive individuals (n = 40), 5.0% were <25 years, 75.0% were 25–50 years, and 20.0% were >50 years. In the negative group (n = 75), 9.3% were <25 years, 70.7% were 25–50 years, and 20.0% were >50 years. No significant difference in age distribution was found between groups (p = 0.707, Chi-squared test). Regarding sex, 22.5% of positive individuals were male and 77.5% female, while in the negative group, 26.7% were male and 73.3% female. No significant differences in sex distribution were observed (p = 0.624, Chi-squared test).
This revision is located in the Results section, subsection 3.1, page 4, lines 172–178. We believe this version enhances the flow of the text and aligns with the reviewer's recommendation for conciseness.

Comments 20:
"Line 193–194: The statement regarding the 'reported at least one comorbidity' is confusing when using the absolute number; c.f. 16 vs 21. I suggest using the percentage unless the denominator is included in the statement, or the absolute values be preceded by the percentage e.g., 40% (21/xxx) compared to 28% (21/XX)."

Response 20:
Thank you for this clear and helpful suggestion. We agree that expressing both the percentage and the absolute values improves clarity and readability. Accordingly, we revised the sentence to explicitly state both forms of the data. The updated version now reads:
Conversely, a statistically significant difference was observed in the prevalence of self-reported comorbidities. Among COVID-19 positive individuals, 40.0% (16/40) reported comorbidities, compared with 28.0% (21/75) in the negative group (p = 0.029, chi-square test).
This change can be found in the Results section, subsection 3.1: Demographic characterization and diagnostic confirmation, lines 179–181, page 4.
We believe this adjustment clarifies the comparison and aligns with the reviewer’s recommendation.

Comments 21:
“Conclusion: The conclusion does not address the core of the study such that the vaccine effect is appearing when this was not adequately explained in the whole write up.”

Response 21:
Thank you for your thoughtful feedback. We appreciate your attention to the coherence between the conclusion and the body of the manuscript. We believe that our conclusion was aligned with the central aims of the study, by highlighting the key clinical and immunological findings derived from the evaluation of individuals infected with SARS-CoV-2 Omicron subvariants.

However, we understand that the inclusion of remarks regarding vaccination—although based on an observed variable in our cohort, could divert attention from the main focus of the manuscript, especially since vaccination effects were not explored in depth throughout the study. In light of this, we revised the Conclusion to maintain a tighter focus on the core findings. Specifically, we removed the sentence referring to vaccine-related aspects.

The revised Conclusion now reads:
This study presents a characterized panel of cytokines related to Th1, Th2, and Th17 pathways, along with the clinical profile of SARS-CoV-2 Omicron subvariant infections. Our findings highlight a high burden of upper respiratory symptoms, consistent with the clinical profile of Omicron-dominant waves, and demonstrate elevated concentrations of Th1-, Th2-, and Th17-associated cytokines in infected individuals compared to negative individuals. Despite the genetic diversity among subvariants, symptom frequency and cytokine levels were not significantly associated with specific lineages, suggesting a conserved clinical and immunological response pattern within the Omicron lineage.
Additionally, a comparative analysis of cytokine profiles between individuals infected with early versus later Omicron subvariants revealed distinct inflammatory patterns, reinforcing the importance of monitoring viral evolution and its immunological implications.
Together, these findings contribute to a growing body of evidence regarding the immune response to Omicron and provide a valuable reference for interpreting responses to currently circulating and future SARS-CoV-2 variants. They underscore the need for ongoing genomic and immunological surveillance, particularly in underrepresented populations, to guide public health strategies and inform targeted interventions.
This change was implemented in the Conclusions section, pages 16, lines 524–540.

A DOCX file with all the changes made to our manuscript has been attached, to better clarify and convey the submission process of our article.

We believe this revision helps ensure consistency and maintains the integrity and focus of the final message.

Reviewer 3 Report

Comments and Suggestions for Authors

I want to thank you for your work. It is a good work to understand the immune responses of Sars CoV 2 but the sample size is very small especially small sample size of subgroups reduces power. Assessment of viral loads may help better interpret cytokine differences. And it may be better to use multivariable regression models to adjust for confounding factors such as age, sex, vaccination, and comorbidities. Symptoms were self-reported and not clinically confirmed or measured in severity.

Author Response

Comment 1:

I want to thank you for your work. It is a good work to understand the immune responses of Sars CoV 2 but the sample size is very small especially small sample size of subgroups reduces power.

Response 1:

Thank you for your kind words and for highlighting this important point. We fully agree with your observation. To address your concern, we have expanded the discussion of our study’s limitations, with a specific mention of the reduced statistical power due to the relatively small sample size and subgroup distribution. This update has been incorporated into the final paragraph of the discussion section (Topic 4, pages 15–16, lines 510–522), which now reads:

Lastly, although the current study offers valuable insights, it is essential to acknowledge several limitations. These include the relatively small sample size, particularly in subgroup analyses, which may limit the statistical power to detect subtle differences; the cross-sectional design, which does not capture temporal changes in cytokine profiles; and the absence of functional immune assays, which could have provided mechanistic insights into the observed cytokine patterns. Additionally, the study did not perform viral co-infection screening in RT-qPCR–negative individuals, which could have further clarified differential immune activation profiles. Despite these limitations, this study contributes important data on the immunological and clinical characterization of SARS-CoV-2 Omicron subvariants in a Brazilian Amazonian population, a region underrepresented in global COVID-19 research. These findings may help inform public health decisions and provide a foundation for future studies seeking to understand immune responses to emerging variants in diverse demographic and immunogenetic backgrounds.

We hope this clarification appropriately addresses your concern and improves the manuscript’s transparency regarding its limitations.

Comment 2:
Assessment of viral loads may help better interpret cytokine differences.

Response 2:
Thank you for this thoughtful and valuable suggestion. We agree that incorporating viral load assessments could provide important context for interpreting cytokine profiles and further elucidate the dynamics of immune activation. Although this analysis was not within the scope of the present study, which primarily focused on diagnostic confirmation and subvariant genotyping for descriptive and comparative purposes, we recognize its relevance and are considering it for future investigations.   

Comment 3:
And it may be better to use multivariable regression models to adjust for confounding factors such as age, sex, vaccination, and comorbidities.

Response 3:
Thank you very much for this valuable suggestion. We fully agree that multivariable regression models are powerful tools to control for potential confounders such as age, sex, vaccination status, and comorbidities. We carefully considered this approach; however, given the relatively small sample size, particularly within certain subgroups, we determined that including multiple covariates in regression models would reduce statistical power and potentially yield unreliable estimates. For this reason, we opted for stratified analyses and non-parametric tests to explore associations. We recognize the importance of adjusting for confounders and acknowledge this as a limitation of our study, which we plan to address in future investigations with larger cohorts. We appreciate your insight and believe it strengthens the interpretation and future direction of our work.

Comment 4:
Symptoms were self-reported and not clinically confirmed or measured in severity.

Response 4:
Thank you for your thoughtful observation. We appreciate the opportunity to clarify this aspect of our study. All individuals included in the cohort sought medical care at outpatient units presenting symptoms compatible with a viral respiratory syndrome. Before inclusion in the study, each participant was initially evaluated by a clinical care team composed of nurses and physicians. Upon identification of a potential COVID-19 case, our research team was then notified and initiated the research procedures, which included ethical clarification, obtaining informed consent, and collecting both sociodemographic/clinical data and biological samples.

To improve clarity and address this point, we revised the description in the “Materials and Methods” section (Section 2.1, page 3, lines 100-103), which now reads:
"This study included biological samples collected from 115 individuals (≥18 years), of both sexes, who sought care for suspected COVID-19 at two outpatient medical units located in th metropolitan region of Belém, Pará, Brazil. Sample collection was conducted between December 2021 and March 2022."

We hope this clarification adequately addresses your concern. 

A DOCX file with all the changes made to our manuscript has been attached, to better clarify and convey the submission process of our article.

Reviewer 4 Report

Comments and Suggestions for Authors

The authors are encouraged to add a sentence or two indicating that the "negative" group likely have some (viral) infection.  Note: the two groups are "confirmed" SARS-CoV-2 group and "other" group with unknown causative agent. (line about 183)

Hyphenation errors in method section:

    line 104 - con-taining

    line 105 - nasopha-ryngeal

    line 106 - se-quencing

    line 110 Nasopharynge-al

Line 123-124 "curated using specialized bioinformatics software" - please be specific (even naming customized tools)

Important errors in data (line 197) RT-qPCR Negative n=75 (%) values are calculated incorrected for most of Table 1!  These must be corrected - example Pharyngitis value 36 of 75 is not 60.0%! etc. etc. etc.

line 202 - this this the chi square 2x2 test?  Please document clearly.

Figure 1B - why is the "Not reported" included in the chart and colored?  Why is not a simple bar/column chart?

Table 2 values - these look like p-values?  Please clarify if multiple testing corrections were or were not applied?  If not, why not?  Other than most of the values were not significant.  My point is that it appears that a very large number of statistical tests are being considered without multiple testing corrections. (see line 498)

Section 3.11 - it would be interesting to have details of which symptoms most commonly represented in the fewer than eight symptoms group and the eight or more symptoms group.

Figure 7 - this manuscript may be improved if the were patterns in the symptoms being represented in the <8 and >=8 groups in the context of 2 doses or 3 doses of vaccine?  If there are no patterns, it would be interesting if the authors would indicate this.

line 474 - please define what are "non-severe cases, acute COVID-19" - this description appears to be at odds with itself

line 547-549 - If the population is receiving COVID-19 vaccines and booster doses but are still getting infected with SARS-CoV-2, it makes this reviewer wonder if there is any "real" efficacy to these vaccines and booster doses?  Reducing symptoms load is nice, but there appears to be no real protection from these COVID-19 vaccines and boosters.  (Actually this reviewer thinks there is only short-term protection only while antibody titer doses are very very high - as soon as Ab titers drop - the individual is again at risk for COVID-19).

line 568 - "comprehensive immunological and clinical assessment" seems to be over stated - characterized panel of cytokines seems to be more accurate - please rephrase this sentence.

line 577 - "protective immunomodulatory role of booster immunization" hasn't been demonstrated - the actual opposite was illustrated with infected vaccinated individuals.  It is fine to state reduced symptoms, but this doesn't really seem to be protection to this reviewer.

Author Response

Comments 1:
"The authors are encouraged to add a sentence or two indicating that the 'negative' group likely have some (viral) infection."

Response 1:
Thank you for this insightful suggestion. We agree that it is important to clarify the definition of the “negative” group. In our study, we carefully designated this group as “negative” based solely on confirmed negative results for SARS-CoV-2 by RT-qPCR. Although these individuals presented symptoms compatible with viral infection, no additional investigation was conducted to determine the specific etiology of those symptoms. Therefore, we used the term “negative” strictly to indicate negativity for SARS-CoV-2, as this was the central focus of our cohort and diagnostic testing.

To make this point clearer to readers, we revised the beginning of subsection 3.1 in the Results to explicitly state that the remaining individuals were negative specifically for SARS-CoV-2. The updated sentence now reads: “A total of 115 adult individuals, aged 18 years or older, participated in this study. Of these, 40 participants (34.78%) were diagnosed with COVID-19 through a positive molecular RT-qPCR test for SARS-CoV-2, while the remaining 75 individuals (65.22%) tested negative for this virus.” This change can be found in the Results section, subsection 3.1, page 4, lines 180–183.

Comments 2:
"Hyphenation errors in method section:
    line 104 - con-taining
    line 105 - nasopha-ryngeal
    line 106 - se-quencing
    line 110 - Nasopharynge-al"

Response 2:
Thank you for carefully reviewing our manuscript and pointing out these hyphenation errors. We have corrected all the indicated terms to ensure proper formatting and clarity. 

Comment 3:
"Line 123–124: "curated using specialized bioinformatics software" – please be specific (even naming customized tools)."

Response:
We appreciate the reviewer’s observation. In response to this comment, we revised the sentence to specify the tools used for data processing. The genomic analysis and variant classification were performed using the ViralFlow pipeline, developed for SARS-CoV-2 genomic surveillance in Brazil. This workflow includes automated preprocessing of sequencing reads, genome assembly, and lineage assignment through the PANGOLIN tool, in addition to Nextclade, which was used for phylogenetic classification and sequence quality assessment.

The revised sentence has been included in the manuscript in section 2.2 Genotyping Data Acquisition (lines 127–132):

"The genomic analysis and classification of SARS-CoV-2 lineages and sublineages were performed using the ViralFlow workflow [15,16]. This pipeline allows for automated processing of sequencing data, including quality control, genome assembly, and lineage assignment. Consensus sequences were generated from high-quality reads and classified using the PANGOLIN tool. Clade assignment and sequence quality were further evaluated using the Nextclade platform." (Section 2. Results, Topic 2.2 Genotyping Data Acquisition)

Comments 4:
"Important errors in data (line 197) RT-qPCR Negative n=75 (%) values are calculated incorrected for most of Table 1! These must be corrected - example Pharyngitis value 36 of 75 is not 60.0%! etc. etc. etc."

Response 4:
We thank the reviewer for the observation. The percentage values in the RT-qPCR negative group of Table 1 have been corrected in the revised manuscript. The changes are reflected in Table 1 and in the Results section, subsection 3.2 (pages 5–6, lines 16-201)

Comment 5:
Line 202: Is this the Chi-square 2x2 test? Please document clearly.

Response 5:
We thank the reviewer for the observation. Yes, this analysis was conducted using a 2x2 Chi-square (χ²) test. To make this clearer for the reader, we have revised the sentence to explicitly include the symbol "(χ²)", as follows:

"The comparison between the groups with positive and negative molecular test results for SARS-CoV-2 revealed a statistically significant difference for some reported symptoms, as determined by the Chi-squared test (χ²) in Table 1."

This change has been incorporated into the Results section, subsection 3.2 Symptomatologic analysis.

Comment 6:
Figure 1B: Why is the "Not reported" included in the chart and colored? Why is it not a simple bar/column chart?

Response 6:
We thank the reviewer for this valuable clarification. We agree that a simple bar chart would more effectively represent the data we intended to convey to the reader, specifically, the frequency distribution of each reported symptom in our SARS-CoV-2 positive group. Based on this suggestion, we have modified the layout of Figure 1B to present the data as a standard bar chart and removed the "Not reported" category from the colored data segments to enhance clarity and interpretation.

This update has been implemented in the revised version of the manuscript.

Comment 7:
Table 2 values: These look like p-values? Please clarify if multiple testing corrections were or were not applied? If not, why not? Other than most of the values were not significant. My point is that it appears that a very large number of statistical tests are being considered without multiple testing corrections. (see line 498)

Response 7:
We thank the reviewer for this important observation. We chose to include the p-values from the binomial logistic regression tests in Table 2 to transparently present the outcome of each symptom–cytokine association to the reader. As correctly noted, a large number of statistical comparisons were performed, and to address this, we applied the Benjamini-Hochberg correction method for multiple testing. Although the nominal p-value for the association between loss of taste and IFN-γ levels was 0.038, this association did not remain significant after correction (adjusted p = 0.955).

This clarification has now been explicitly added to the manuscript (page 8–9, lines 293–308).

[Updated text in the manuscript:]
"Only one comparison yielded a nominal p-value below the conventional significance threshold: the association between loss of taste and IFN-γ levels (p = 0.038). However, this result did not remain significant after correction for multiple comparisons using the Benjamini-Hochberg method (adjusted p = 0.955)."

Comment 8: Figure 7 - this manuscript may be improved if the were patterns in the symptoms being represented in the <8 and >=8 groups in the context of 2 doses or 3 doses of vaccine? If there are no patterns, it would be interesting if the authors would indicate this.

Response 8:
We appreciate this insightful analytical suggestion. While we recognize the relevance of exploring symptom patterns in the context of vaccination doses, we chose not to delve into this specific analysis to maintain focus on the central objectives of our study. The included analysis was motivated by an internal question within our group about the potential influence of vaccine doses on the symptom burden and, consequently, on the clinical outcomes of volunteers participating in our study.

Cooment 9: "line 474 - please define what are "non-severe cases, acute COVID-19" - this description appears to be at odds with it self"

Response 9: We thank the reviewer for this valuable observation and we fully agree that the original phrasing lacked clarity. To better convey our intended meaning, we have revised the sentence in the manuscript to read:

These findings reinforce the notion that, even in mild-to-moderate cases during the acute phase of COVID-19, a broad symptom spectrum can be observed, often overlapping with other respiratory infections. This highlights the importance of molecular diagnostics for accurate etiological confirmation.” (Section 4, Discussion, paragraph 2, lines 419-423)

This revision aims to eliminate any contradiction and more accurately reflect the clinical context addressed in our study.

Comment 10: "line 547-549 - If the population is receiving COVID-19 vaccines and booster doses but are still getting infected with SARS-CoV-2, it makes this reviewer wonder if there is any "real" efficacy to these vaccines and booster doses?  Reducing symptoms load is nice, but there appears to be no real protection from these COVID-19 vaccines and boosters.  (Actually this reviewer thinks there is only short-term protection only while antibody titer doses are very very high - as soon as Ab titers drop - the individual is again at risk for COVID-19)."

Response 10: We thank the reviewer for this thoughtful comment. In our study, vaccination status was indeed recorded; however, this parameter was based on self-reported data and not subsequently validated through immunological assays, such as quantification of neutralizing antibodies or evaluation of cellular immune responses. Therefore, we chose not to establish direct correlations between vaccination and clinical or immunological outcomes, as doing so could introduce bias or lead to overinterpretation. To better reflect this decision and avoid misinterpretation, we have removed the phrase “On vaccine-induced immunomodulation” from Section 4, paragraph 12, line 671. 

Comment 11:
Line 568 – “comprehensive immunological and clinical assessment” seems to be overstated – characterized panel of cytokines seems to be more accurate – please rephrase this sentence.

Response 11:
We thank the reviewer for the suggestion and agree with the observation. The sentence has been revised to:
"This study presents a characterized panel of cytokines related to Th1, Th2, and Th17 pathways, along with the clinical profile of SARS-CoV-2 Omicron subvariant infections."
(Section 5, Conclusions, first paragraph)

Comment 12: "line 577 - "protective immunomodulatory role of booster immunization" hasn't been demonstrated - the actual opposite was illustrated with infected vaccinated individuals.  It is fine to state reduced symptoms, but this doesn't really seem to be protection to this reviewer."

Response 12: We thank the reviewer for this observation. We agree that the sentence may have overstated our findings. As such, we have removed this statement from the Conclusions section to ensure that our conclusions remain aligned with the core focus of the study.

We appreciate your valuable comments and insights into our work. A DOCX file with all the changes made to our manuscript has been attached to better clarify and convey the submission process.

Round 2

Reviewer 2 Report

Comments and Suggestions for Authors

Dear Authors

Thank you for taking time to consider the suggestion and explaining the areas that needed clarification.

Please consider the nomenclature for Chi-Square in the document.

Thank you.

Author Response

Dear Reviewer,

We would like to sincerely thank you for your dedication and valuable contributions to our manuscript. Your detailed comments and constructive suggestions were essential in improving the quality of our work. We greatly appreciate the time and effort you invested throughout the review process.

Comment:"Please consider the nomenclature for Chi-Square in the document."

Response: We thank the reviewer for this observation. Following the suggestion, we carefully revised the manuscript and updated the nomenclature by standardizing the term to "Chi-Square" throughout the document.

Reviewer 3 Report

Comments and Suggestions for Authors

Subgroup analysis by omicron subvariant is limited by small sample sizes, which limits statistical power. Authors should clarify this in the limitations section. When evaluating symptom profiles, the clinical significance of cytokine elevations (e.g., predictive value, relationship to severity) can be discussed in more depth. The conclusion that omicron subvariants do not drive distinct cytokine profiles is important. However, the discussion needs to further explore the possible immunological or virological mechanisms underlying this homogeneity.

Comments on the Quality of English Language

The manuscript requires careful editing. Numerous minor typographical and grammatical errors (e.g., "has been done," "has been done with due care") compromise readability. Some paragraphs and sentences appear repeated or fragmented (especially in the abstract and introduction); these should be combined or removed.

Author Response

Dear Reviewer,

We would like to sincerely thank you for your dedication and valuable contributions to our manuscript. Your detailed comments and constructive suggestions were essential in improving the quality of our work. We greatly appreciate the time and effort you invested throughout the review process.

Comment 1:"Subgroup analysis by omicron subvariant is limited by small sample sizes, which limits statistical power. Authors should clarify this in the limitations section."

Response 1: We thank the reviewer for this important observation. Following the suggestion, we revised the Discussion section to explicitly address this limitation. The manuscript now states:

"Lastly, although the present study provides valuable insights, several limitations must be acknowledged. These include the relatively small sample size, which is particularly restrictive in subgroup analyses by Omicron subvariant, thereby limiting the statistical power to detect subtle differences; the cross-sectional design, which does not capture temporal changes in cytokine profiles; and the absence of functional immunological assays, which could have provided mechanistic insights into the observed cytokine patterns. Moreover, the study did not perform viral coinfection screening in RT-qPCR–negative individuals, which might have further clarified differential immune activation profiles." (Section 4. Discussion, paragraph 15, lines 507–514) 

Comment 2: "When evaluating symptom profiles, the clinical significance of cytokine elevations (e.g., predictive value, relationship to severity) can be discussed in more depth. "

Response 2: We thank the reviewer for this important comment. We fully agree that discussing the clinical significance of cytokine elevations, particularly in terms of predictive value and relationship to severity, is highly relevant. However, our study was designed with a cross-sectional, outpatient approach focusing on the quantification of Th1-, Th2-, and Th17-associated cytokines in individuals infected with Omicron subvariants. The study population was recruited from an ambulatory setting, and no longitudinal follow-up or physician-based clinical severity assessment was available. Therefore, our dataset does not allow us to directly evaluate the predictive value of cytokines or their association with disease severity.

Nonetheless, we sought to address the reviewer’s point by contextualizing our findings with respect to previous literature. Several sections of the Discussion highlight that, although SARS-CoV-2–positive individuals displayed elevated cytokine levels compared to negatives, the overall inflammatory profile was more controlled than that described during earlier pandemic waves, when cytokine storms were key drivers of severity and mortality. For instance:

"The absence of statistically significant differences in IL-6 and IL-17A may suggest a more controlled inflammatory response in Omicron infections compared to earlier variants, consistent with the clinical observation of reduced severity during Omicron-dominant waves" (Section 4, Discussion, lines: 448-451).

"The cytokine correlation matrix revealed… a dynamic interplay between effector and regulatory cytokines… rather than a dysregulated cytokine storm" (Section 4, Discussion, lines: 481-494).

"Such a balanced inflammatory environment is particularly relevant in the context of Omicron-dominant infections… offering immunological insight into the mechanisms that contributed to reduced hospitalization rates despite high transmission levels" (Section 4, Discussion, lines: 495-500).

Together, these statements emphasize that, while cytokine elevations were present in Omicron infections, they appear less extreme than in earlier variants and may contribute to explaining the milder clinical presentations generally reported. We also clearly acknowledge the limitation that our study does not allow direct inference of predictive or prognostic value, and that such analyses would require longitudinal cohorts with clinical outcome data

Comment 3: "The conclusion that omicron subvariants do not drive distinct cytokine profiles is important."

Response 3: We thank the reviewer for this observation and fully agree. As noted in our conclusion, in our study, Omicron subvariants did not show differences in the quantification of Th1-, Th2-, and Th17-associated cytokines. "Despite the genetic diversity among subvariants, symptom frequency and cytokine levels were not significantly associated with specific lineages, suggesting a conserved clinical and immunological response pattern within the Omicron lineage." [Section 5.Conclusions, lines: 538-541]. 

Comment 4: "However, the discussion needs to further explore the possible immunological or virological mechanisms underlying this homogeneity."

Response 4: We thank the reviewer for this valuable suggestion. We agree that discussing the possible immunological and virological mechanisms underlying the homogeneity observed in our cohort is important. Our Discussion already touches on these aspects in several passages. For example, we noted that "this observation could be attributed to functional conservation in viral entry or replication mechanisms across subvariants, or alternatively, to host-related factors such as immune status, vaccination history, or comorbidities" (Section 4, Discussion, lines: 436-438). We also emphasized that "the systemic pro-inflammatory immune profile remains relatively conserved across circulating subvariants, supporting the hypothesis of a more uniform immune response within this lineage’s evolutionary trajectory" (Section 4, Discussion, lines: 470-472).

In addition, following the reviewer’s suggestion, we have expanded this point by adding the following sentence: "This homogeneity may also reflect the strong selective pressure shaping Omicron’s evolution, favoring subvariants with similar host–virus interaction dynamics despite accumulating mutations in the spike protein." (Section 4, lines 472–474).”

Comment 5: "The manuscript requires careful editing. Numerous minor typographical and grammatical errors (e.g., "has been done," "has been done with due care") compromise readability. Some paragraphs and sentences appear repeated or fragmented (especially in the abstract and introduction); these should be combined or removed."

Response 5: We thank the reviewer for this careful observation regarding typographical and grammatical issues in the manuscript. We fully agree that these details are essential for improving readability and clarity. In response, we performed a thorough review of the entire text, including the abstract and introduction, to identify and correct typographical errors, fragmented or repeated sentences, and inconsistencies in grammar and style. The revised version has been carefully edited to ensure accuracy, conciseness, and fluency.

Reviewer 4 Report

Comments and Suggestions for Authors

Repeat request: The authors are encouraged to add a sentence or two indicating that the 'negative' group likely have some unconfirmed (viral) infection."

Deleted lines 130-132 remove method details that are useful for documenting the methods used.  If they were not added in a different section, the authors are encouraged to consider restoring these details.

Repeat request: multiple percentages in Table 1 are not calculated correctly: 

  • Age ranges
  • Sex

Table 1 p-values appear to be uncorrected for multiple testing.  Please document clearly if these p-values are corrected or uncorrected for multiple testing.  

  • The p-value of 0.041 for Painful breathing may not remain significant after multiple testing correction.  This may change statement(s) in the manuscript.

Line 237 seems overstated - details presented on line 243-244.  For only (2?) symptoms (pharyngitis and chills) of the list seems a bit at odds with the claim on line 237 (not incorrect, but overstated).

Repeat request: Are the values presented in Table 2 corrected or uncorrected for multiple testing.  Please document clearly.

Line 581 seems at odds with the results in Table 1 (pharyngitis and chills being the main difference).  Please cite specific differences with data references to support this claim.

The authors are encouraged to recheck calculations presented in the article.

Author Response

Dear Reviewer,

We would like to sincerely thank you for your dedication and valuable contributions to our manuscript. Your detailed comments and constructive suggestions were essential in improving the quality of our work. We greatly appreciate the time and effort you invested throughout the review process.

Comment 1:"The authors are encouraged to add a sentence or two indicating that the 'negative' group likely have some unconfirmed (viral) infection."

Response  2: We thank the reviewer for this valuable suggestion. We agree with the point raised and have incorporated a more descriptive statement to acknowledge this limitation in our manuscript. The revised text now reads as follows (Section 4. Discussion, lines 512–517):
"Moreover, the study did not perform viral coinfection screening in RT-qPCR–negative individuals, which might have further clarified differential immune activation profiles. It is therefore plausible that some participants classified as SARS-CoV-2–negative were in fact experiencing other unconfirmed viral infections, which may partially explain the clinical manifestations and cytokine responses observed in this group."

Comment 2: "Deleted lines 130-132 remove method details that are useful for documenting the methods used.  If they were not added in a different section, the authors are encouraged to consider restoring these details."

Response 2: We thank the reviewer for this helpful comment. We initially removed the detailed description of sequencing steps in order to improve the cohesion and readability of the Methods section, since our genomic analyses followed the standardized and peer-reviewed ViralFlow workflow [15,16], which is widely used in genomic surveillance in Brazil and referenced by GISAID. As such, we considered that citing the published workflow would be sufficient to guide readers interested in the full methodological details.

However, we agree with the reviewer that briefly restoring some descriptive steps enhances methodological transparency. We have therefore reintroduced a concise description of the essential processes while keeping the reference to ViralFlow. The revised text now reads as follows (Materials and Methods; subsection 2.2 Genotyping Data Acquisition, second paragraph, lines 121–127):

"The genomic analysis and classification of SARS-CoV-2 lineages and sublineages were performed using the ViralFlow workflow [15,16]. Raw sequencing reads underwent quality control and were aligned to the SARS-CoV-2 reference genome (NC_045512.2) as part of this pipeline. This workflow allows for automated processing of sequencing data, including genome assembly and lineage assignment. Consensus sequences were generated from high-quality reads and classified using the PANGOLIN tool. Clade assignment and sequence quality were further evaluated using the Nextclade platform".

Comment 3: "Repeat request: multiple percentages in Table 1 are not calculated correctly: Age ranges; Sex"

Response 3: We thank the reviewer for this sharp observation. After carefully reviewing Table 1, we identified that the percentages were originally calculated in relation to the total study population (i.e., combining SARS-CoV-2–positive and –negative groups), rather than within each group separately. For example, in the variable Sex, the percentage of males was calculated based on the total sample size instead of the group-specific denominators. We agree that this presentation could be misleading. The values have now been corrected to reflect the proper within-group percentages. The revised and accurate data are presented in Table 1 of the manuscript.

Comment 4: "Table 1 p-values appear to be uncorrected for multiple testing.  Please document clearly if these p-values are corrected or uncorrected for multiple testing."

Response 4: We thank the reviewer for this important observation regarding multiple testing correction. We have now applied the Benjamini–Hochberg false discovery rate (FDR) method to adjust the p-values across the 16 symptoms analyzed. After correction, only chills remained statistically significant between groups (adjusted p = 0.014). Symptoms such as pharyngitis (uncorrected p = 0.017; adjusted p = 0.136) and painful breathing (uncorrected p = 0.041; adjusted p = 0.212), which initially reached significance, did not remain significant after adjustment. All other symptoms showed no statistically significant differences (p > 0.05).

These corrections have been implemented in Table 1 and are now clearly documented in the Results section (Section 3.2 Symptomatologic analysis; lines 192–198), which reads as follows:

After correction for multiple comparisons using the Benjamini-Hochberg false discovery rate (FDR) method, a statistically significant difference was observed between the groups only for chills (adjusted p = 0.014). Pharyngitis (before adjusted in p = 0.017 × adjusted p = 0.136) and painful breathing (before adjusted in p = 0.041 × adjusted p = 0.212), which were initially significant before correction, were not significant after adjustment. The other symptoms did not show statistically significant differences between the groups (p > 0.05).”

This adjustment ensures proper statistical rigor and transparency in reporting our findings

Comment 5: "The p-value of 0.041 for Painful breathing may not remain significant after multiple testing correction.  This may change statement(s) in the manuscript."

Response 5: We thank the reviewer for this valuable observation. We agree that the previously reported p-value for painful breathing (0.041) does not remain significant after correction for multiple comparisons. Accordingly, we have updated the description in the manuscript to reflect the corrected data. The revised text now reads as follows (Section 3.2 Symptomatologic analysis; lines 195–197):

"Pharyngitis (before adjustment p = 0.017 × adjusted p = 0.136) and painful breathing (before adjustment p = 0.041 × adjusted p = 0.212), which were initially significant before correction, were not significant after adjustment."

Comment 6: "Line 237 seems overstated - details presented on line 243-244.  For only (2?) symptoms (pharyngitis and chills) of the list seems a bit at odds with the claim on line 237 (not incorrect, but overstated)."

Response 6: We thank the reviewer for this observation. We agree that the original statement on line 237 was somewhat overstated given that only two symptoms (pharyngitis and chills) showed notable differences. Accordingly, we have revised the text to more accurately reflect our analysis. The updated passage now reads as follows (Section 3. Results, subsection 3.4 Correlation of the most frequent symptoms of the positive group with the identified variants; lines 237–241):

"To explore whether the occurrence of certain clinical symptoms in SARS-CoV-2–positive individuals differed according to Omicron subvariants, we conducted Fisher’s exact test to evaluate the relationship between each of the four most frequently reported symptoms, pharyngitis, nasal discharge, headache, and cough, and the 11 Omicron subvariants identified in the cohort (n = 40)."

Comment 7: "Repeat request: Are the values presented in Table 2 corrected or uncorrected for multiple testing.  Please document clearly."

Response 7: We thank the reviewer for this repeated and important observation. We confirm that all p-values presented in the logistic regression analysis were corrected for multiple testing using the Benjamini–Hochberg (BH) method to control the false discovery rate (FDR).

Upon internal discussion, we also recognized that the original presentation of these data within the main text could potentially cause confusion for readers due to the volume of p-values reported. To improve clarity, we decided to move the full tables (including both nominal and adjusted p-values) to the Supplementary Material. This provides transparency while avoiding overloading the main manuscript. The supplementary dataset has been deposited in Zenodo (DOI: https://doi.org/10.5281/zenodo.16923060
).

The revised text in the Results section now reads as follows (Section 3. Results, subsection 3.6 Association between clinical symptoms and plasma cytokine levels in patients with acute COVID-19; lines 275–292):

"To investigate whether specific clinical symptoms were associated with alterations in cytokine levels, we conducted an association analysis between the 16 self-reported symptoms and the plasma concentrations of seven cytokines: IL-17A, IFN-γ, TNF, IL-10, IL-6, IL-2, and IL-4, in individuals who tested positive for SARS-CoV-2. The corresponding p-values for each cytokine–symptom comparison are presented in Table 1 of the Supplementary Material. Among the comparisons performed, two nominal p-values were below the conventional significance threshold: the association between taste loss and IFN-γ levels (p = 0.038) and abdominal pain and TNF levels (p = 0.026). However, these results did not remain significant after correction for multiple comparisons using the Benjamini–Hochberg method (adjusted p = 0.962 for both). Post-correction p-values are presented in Table 2 of the Supplementary Material. All other symptoms, including pharyngitis, headache, fever, dyspnea, myalgia, and others, showed no statistically significant associations with any of the measured cytokines. These findings suggest that, in this cohort, individual clinical manifestations were not significantly associated with circulating levels of Th1-, Th2-, or Th17-associated cytokines, nor with the central inflammatory marker IL-6."

Comment 8: "Line 581 seems at odds with the results in Table 1 (pharyngitis and chills being the main difference).  Please cite specific differences with data references to support this claim."

Response 8: We thank the reviewer for this insightful comment. We agree that our original wording overstated the clinical differences between groups and did not accurately reflect the results shown in Table 1. In response, we have revised this section of the Discussion to better align with our findings and the existing literature. The new text emphasizes the distinct clinical profile of Omicron infections compared with earlier pandemic waves, highlights the lower frequency of hallmark symptoms such as loss of smell and taste, and discusses the overall higher symptom burden observed in the SARS-CoV-2–positive group. At the same time, it acknowledges the overlap of symptoms between positive and negative individuals and underscores the importance of molecular diagnostics due to the lack of specificity in clinical presentation.

The revised passage now reads as follows (Section 4. Discussion, lines 418-431):

"Compared to the aggressive and characteristic clinical profile of COVID-19 observed during the first pandemic waves, infections caused by the Omicron variant presented a dis-tinct symptom pattern in our cohort. Symptoms such as loss of smell and taste, previously considered hallmarks of SARS-CoV-2 infection, were reported at much lower frequencies, consistent with previously published studies on Omicron waves. This change reflects the reduction in clinical severity described for Omicron, although vulnerable groups remain at risk of severe outcomes [18].
Our analysis also demonstrated that, despite the specific symptom overlap between SARS-CoV-2-positive and -negative individuals, the positive group reported a higher over-all symptom burden. This reinforces the notion that, although disease severity has de-creased compared to previous variants, as discussed elsewhere, Omicron infections still present a broad clinical spectrum that distinguishes them from other respiratory infections [19-20]. Furthermore, the lack of specificity in clinical presentation highlights the essential role of molecular laboratory diagnosis for the accurate identification of COVID-19."

Round 3

Reviewer 3 Report

Comments and Suggestions for Authors

Your manuscript provides valuable and timely insights into the cytokine responses associated with SARS-CoV-2 Omicron subvariants and their clinical correlations. The study is well executed, and the findings contribute meaningfully to understanding immune responses during Omicron infections. Thank you for your work.

Author Response

We sincerely thank the reviewer for the generous and encouraging feedback. We greatly appreciate the time and effort dedicated to evaluating our manuscript and are pleased that our study is recognized as a valuable contribution to the understanding of cytokine responses and clinical correlations in the context of SARS-CoV-2 Omicron infections.

Reviewer 4 Report

Comments and Suggestions for Authors

This revision is an improvement over the previous version.

Line 52 - "non-infected controls" should be "non-SARS-CoV-2 infected controls" - this individuals likely have some unknown infections; they do not represent a negative control group - they represent an infected non-COVID-19 comparison group.

Line 101-102 has unusual wording.  Preprints and later publications predicted multiple variants for the SARS-CoV-2 Spike protein.

Line 287, 300, 314 - "negative group" should be "negative SARS-CoV-2 group"

Lines 336-338 - Were the p-values being presented corrected for multiple testing?  Please document clearly if these are corrected or uncorrected p-values.

Lines - 406-409 - Sentence structure has capital letters following semicolons - Cough, Nasal, & Headache - these should not be capitalized.

Line 420 - Would be better as the last sentence of the previous paragraph.

Figure 2 - the negative data is difficult to see with the current color selected

Figure 3 - The colors for the bar charts vary but are not documented - what does the differences in colors indicate?

Figure 4 - Like Figure 3, the colors vary without being explained - what do the different colors indicate?

Line 985 - "inform targeted interventions" - this needs to be explained in more detail and justified by the results of this article.

Author Response

We sincerely thank the reviewer for the time, effort, and valuable insights dedicated to evaluating our manuscript. The constructive comments and suggestions have greatly contributed to improving the clarity, accuracy, and overall quality of our work.

Comment 1: "Line 52 - "non-infected controls" should be "non-SARS-CoV-2 infected controls" - this individuals likely have some unknown infections; they do not represent a negative control group - they represent an infected non-COVID-19 comparison group."

Response 1: We thank the reviewer for this pertinent observation. We agree with the suggested terminology and have replaced “non-infected controls” with “non-SARS-CoV-2 infected controls” throughout the manuscript.

Comment 2: Line 101-102 has unusual wording.  Preprints and later publications predicted multiple variants for the SARS-CoV-2 Spike protein.

Response 2: We thank the reviewer for this valuable comment and agree with the observation. To better align the sentence with what was described in the literature regarding the predictions of mutations in the Spike protein, we revised the text in our manuscript, which now reads:
The emergence of multiple SARS-CoV-2 variants, largely driven by mutations in the Spike protein as anticipated by early genomic studies, has significantly shaped the epidemiological trajectory of the COVID-19 pandemic [10,11].”
(Section 1. Introduction, lines 101–103).

Comment 3: "Line 287, 300, 314 - "negative group" should be "negative SARS-CoV-2 group"

Response 3: We thank the reviewer for this helpful suggestion. We revised the manuscript accordingly, replacing the term “negative group” with “negative SARS-CoV-2 group” in all indicated instances (lines 287, 300, and 314)

Comment 4: "Lines 336-338 - Were the p-values being presented corrected for multiple testing?  Please document clearly if these are corrected or uncorrected p-values."

Response 4: We thank the reviewer for raising this important point. As suggested, we have clarified in the manuscript that the reported p-values were not corrected for multiple testing, since the chi-square test was applied only for overall group comparisons rather than pairwise analyses. The revised text now reads:

"Among the 16 symptoms assessed, four showed statistically significant frequencies (uncorrected p < 0.05), indicating higher-than-expected occurrences: pharyngitis (p = 0.017), cough (p = 0.008), nasal discharge (p = 0.014), and headache (p = 0.029). These results were derived from overall comparisons rather than pairwise analyses, and therefore no post hoc correction for multiple testing was applied. These symptoms are highlighted in Figure 1B."
(Section 3, Subsection 3.2 Symptomatologic, lines 335–340).

Comment 5: "Lines - 406-409 - Sentence structure has capital letters following semicolons - Cough, Nasal, & Headache - these should not be capitalized."

Response 5: We thank the reviewer for this sharp observation. We have corrected the grammatical details by changing the capitalization of the terms “cough,” “nasal,” and “headache” following the semicolons in lines 406–409.

Comment 6: "Line 420 - Would be better as the last sentence of the previous paragraph."

Response 6: We thank the reviewer for this helpful suggestion. Following the recommendation, we moved the sentence from line 420 to the end of the previous paragraph to improve the logical flow of the text. This modification appears in Section 3. Results, Subsection 3.4 Correlation of the frequent symptoms of the positive group with the identified variants, lines 243–255.

Comment 7: "Figure 2 - the negative data is difficult to see with the current color selected"

Response 7: We thank the reviewer for pointing out this issue. We have revised Figure 2 by adjusting the color scheme to improve the visibility of the negative data, and we hope that the updated version now meets the expected quality and clarity.

Commet 8 and 9: "Figure 3 - The colors for the bar charts vary but are not documented - what does the differences in colors indicate?" AND "Figure 4 - Like Figure 3, the colors vary without being explained - what do the different colors indicate?"

Response 8 and 9: We thank the reviewer for this observation. The color variations in Figures 3 and 4 were used solely for visual distinction of cytokines within each graph. In Figure 3, each bar chart presents different shades of blue assigned to individual cytokines, and in Figure 4, paired blotplots are similarly distinguished by different shades of blue. These color differences are arbitrary and serve only an aesthetic purpose; they do not convey additional analytical meaning. We also chose to present our graphs using blue-toned palettes to better align with the overall layout of the journal.

Comment 10: "Line 985 - "inform targeted interventions" - this needs to be explained in more detail and justified by the results of this article."

Response 10: We thank the reviewer for this important observation. We agree that the phrase “inform targeted interventions” required further clarification. To avoid ambiguity, we revised the sentence to explicitly connect our findings with their potential implications for public health. The updated version now reads:
Together, these findings contribute to a growing body of evidence regarding the immune response to Omicron and provide a valuable reference for interpreting responses to currently circulating and future SARS-CoV-2 variants. They underscore the need for ongoing genomic and immunological surveillance, particularly in underrepresented populations, to guide public health strategies and support targeted actions such as monitoring symptom burden in vulnerable groups, prioritizing individuals with comorbidities, and optimizing vaccine booster policies.”
(Section 5, Conclusions, page 25, lines 984–989). 
